# Predicting ocean-induced ice-shelf melt rates using deep learning

Sebastian H. R. Rosier[1,2], Christopher Y. S. Bull[1], Wai. L. Woo[3], and G. Hilmar Gudmundsson[1]

[1]Department of Geography and Environmental Sciences, Northumbria University, Newcastle Upon Tyne, UK
[2]WSL Institute for Snow and Avalanche Research SLF, Davos, Switzerland
[3]Department of Computer and Information Sciences, Northumbria University, Newcastle Upon Tyne, UK

**Correspondence:** Sebastian Rosier (sebastian.rosier@northumbria.ac.uk)

**Abstract.** Through their role in buttressing upstream ice flow, Antarctic ice shelves play an important part in regulating future sea level change. Reduction in ice-shelf buttressing caused by increased ocean-induced melt along their undersides is now understood to be one of the key drivers of ice loss from the Antarctic Ice Sheet. However, despite the importance of this forcing mechanism, most ice-sheet simulations currently rely on simple melt-parametrisations of this ocean-driven process since a fully coupled ice-ocean modelling framework is prohibitively computationally expensive. Here, we provide an alternative approach that is able to capture the greatly improved physical description of this process provided by large-scale ocean-circulation models over currently employed melt-parameterisations, but with trivial computational expense. This new method brings together deep learning and physical modelling to develop a deep neural network framework, MELTNET, that can emulate ocean model predictions of sub-ice shelf melt rates. We train MELTNET on synthetic geometries, using the NEMO ocean model as a ground-truth in lieu of observations to provide melt rates both for training and to evaluate the performance of the trained network. We show that MELTNET can accurately predict melt rates for a wide range of complex synthetic geometries, with a normalized root mean squared error of $0.11\,\mathrm{m\,yr^{-1}}$ compared to the ocean model. MELTNET calculates melt rates several orders of magnitude faster than the ocean model and outperforms more traditional parameterisations for >96% of geometries tested. Furthermore, we find MELTNET's melt rate estimates show sensitivity to established physical relationships such as changes in thermal forcing and ice shelf slope. This study demonstrates the potential for a deep learning framework to calculate melt rates with almost no computational expense, that could in the future be used in conjunction with an ice sheet model to provide predictions for large-scale ice sheet models.

## 1 Introduction

Ocean induced melting of ice shelves is currently the main driver of ice mass balance change in Antarctica and represents a major source of uncertainty for predictions of future sea level rise (Pritchard et al., 2012; Gudmundsson et al., 2019; Edwards et al., 2021; De Rydt et al., 2021; IPCC, 2021). Enhanced melting, resulting in a decrease in ice shelf thickness, can lead to a reduction in the buttressing force that ice shelves impose on the ice sheet and thereby a net increase in mass loss (Thomas, 1979; Dupont and Alley, 2005; Gudmundsson et al., 2012). Strong feedbacks between melt rates, the cavity geometry, and ocean circulation can lead to complex temporal and spatial heterogeneity at many different scales, so modelling these processes is challenging (Donat-Magnin et al., 2017; De Rydt et al., 2014; Jordan et al., 2018; Smith et al., 2021; Kreuzer et al., 2021).

The current generation of ice sheet models employ different approaches to deal with this problem and these can broadly be split into three main categories: (1) simple parameterisations that depend on one or more local quantities (e.g. ice shelf draft), (2) intermediate complexity parameterisations that incorporate local and nonlocal processes and include some basic physics of the circulation in an ice shelf cavity (some examples are described later) and (3) ocean general circulation models that are able to capture most of the processes leading to melting at the ice shelf base (Favier et al., 2019). Each of these approaches comes with advantages and disadvantages but broadly speaking the main tradeoff is computational expense vs. fidelity to our best current understanding of ocean circulation within an ice-shelf cavity.

Difficulties in capturing complex physical processes within large scale models is not a problem unique to glaciology. A large source of uncertainty in global climate models running at standard resolution arises from an inability to resolve important cloud processes or accurately calculate convection (Stevens and Bony, 2013). Higher resolution models that do, running at up to 1km resolution, are too expensive to run for longer than one year (e.g. Khairoutdinov and Randall, 2003). A very recent innovation in these fields is to train a neural network, using high resolution models, to parameterise these processes more accurately in a coarser resolution model (e.g. Rasp et al., 2018; O'Gorman and Dwyer, 2018; Brenowitz and Bretherton, 2018, 2019). In this way, the fidelity of physical models running at lower resolution, tractable for long term global prediction, can be greatly improved with almost no additional computational cost.

In a similar vein, a major hurdle for modelling efforts of the Antarctic Ice Sheet is the considerable difference in timescales at which changes occur in the ice sheet and the surrounding ocean. To provide accurate prediction and retain numerical stability, large-scale ocean models typically require timesteps of $\mathcal{O}(10^2 - 10^3 s)$ whereas ice sheet models can often be run at timesteps of $\mathcal{O}(10^7 s)$ or more. This disparity means that simulations that couple an ocean model to an ice sheet model are severely constrained in simulation time and are typically restricted to making predictions for hundreds of years (e.g. Jordan et al., 2018; Thoma et al., 2015; Seroussi et al., 2017; Naughten et al., 2021). That restriction is pertinent, since the full extent of committed sea level rise arising from a changing climate takes millennia to manifest itself in ice sheet models (Garbe et al., 2020). In addition, the computational cost of the ocean models is restrictive and coupled ice-ocean models are often limited to regional simulations, precluding the ability to model how retreat of one basin affects a neighbouring basin. A third and perhaps most important drawback of this high computational cost is that coupled studies typically rely on a smaller set of simulations which cannot properly sample the model parameter space, making it very hard to calculate uncertainty estimates which are now widely recognised as a vital component of predictions.

Despite the numerous drawbacks in using a coupled approach, ocean models present a significant advantage in their ability to reproduce the complex physical processes that lead to melt rate patterns beneath an ice shelf. Capturing the spatial distribution accurately is important because relatively small regions of an ice shelf are disproportionately important to the transient response of ice flow (Goldberg et al., 2019). Simpler parameterisations can be tuned to better match observations to some extent but it is not clear that these tuned models remain valid for longer simulations as the ice sheet geometry and ocean conditions diverge significantly from their present day configuration. Were a paramterisation to exist that did not require tuning, this would represent a major step forward in long-term predictions for the future of the Antarctic ice sheet.

Given the current gulf between ocean models and lower complexity parameterisations often used in ice sheet models, and the aforementioned problems with making long-term forecasts for the Antarctic ice sheet using a fully-coupled approach, there is a clear need for an alternative middle ground. This should retain the ability to predict complex spatial patterns of melting but be computationally efficient in order to be able to run it synchronously with an ice sheet model without inhibiting the size of the domain or the duration of the simulation. Here, we propose using deep learning to emulate ocean model behaviour for the

prediction of sub ice-shelf melt rates. Since the computational cost of a machine learning algorithm is insignificant once it has been trained, this could provide an alternative modelling approach. By treating the ocean model as a ground-truth and running ocean simulations on a wide variety of ice shelf configurations and ocean conditions, a network can be trained to approximate the behaviour of an ocean model. As a first step towards this goal, we demonstrate a deep learning framework that can accurately reproduce melt rate patterns as predicted by the NEMO ocean model and shows significantly better performance than existing

intermediate complexity parameterisations. This predictive ability comes despite a drastically lower computational cost that is not only several orders of magnitude faster than the ocean model it emulates but also faster than the intermediate complexity parameterisations that we have used for comparison.

## 2    Models and Methods

In the absence of sufficiently large observational melt rate training data sets for effective deep learning, we generate random

and synthetic geometries, together with temperature and salinity forcing, for several thousand ice shelves. These inputs are used as forcings for NEMO, a general circulation ocean model, which gives a simulated ice shelf melt rate. The inputs and resulting NEMO melt rates are then applied within our deep learning framework, MELTNET, to train a model that can predict melt rates that closely resemble those predicted by the NEMO ocean model. We follow a standard approach in machine learning in which the inputs are split into training, validation and test sets: The training set is used exclusively to train the network; the

validation set is used to select and optimise model hyperparameters; and the test set, not seen by the network during either training or validation, is used to evaluate the performance of MELTNET and the other melt rate parameterisations selected for comparison.

    We begin by describing the NEMO ocean model, followed by our deep learning methodology. We then explain how the synthetic input fields, consisting of ice shelf geometry, bathymetry, temperature and salinity fields, are generated. Finally, we

introduce the two intermediate complexity melt rate paramterisations, commonly used in the ice sheet modelling community, that we compare MELTNET performance against.

### 2.1    NEMO Ocean modelling

The ocean general circulation model used in this study is version v4.0.4 of the Nucleus for European Modelling of Ocean model (NEMO; Madec and Team). NEMO solves the incompressible, Boussinesq, hydrostatic primitive equations with a split-

explicit free-surface formulation. NEMO here uses a $z^\star$-coordinate (varying cell thickness) C-grid with partial cells at the bottom-most and top-most ocean layers in order to provide more realistic representation of bathymetry (Bernard et al., 2006)

and the ice-shelf geometry, respectively. Our model settings include: a 55-term polynomial approximation of the reference Thermodynamic Equation Of Seawater (TEOS-10; IOC and IAPSO (2010)), nonlinear bottom friction, a free-slip condition at the lateral boundaries (at both land and ice shelf interfaces), energy- and enstrophy-conserving momentum advection scheme and a prognostic turbulent kinetic energy scheme for vertical mixing (Madec et al., 1998). Laterally, we have spatially varying eddy coefficients (according to local mesh size) with a Laplacian operator for iso-neutral diffusion of tracers and a biharmonic operator for lateral diffusion of momentum. Our model setup utilises the ice-shelf module that was developed by Mathiot et al. (2017). Calculation of the ice shelf melt rate follows the standard three-equation parameterization as described in Asay-Davis et al. (2016), with heat exchange and salt exchange coefficients of $\Gamma_T = 6 \times 10^{-2}$ and $\Gamma_s = 1.7 \times 10^{-3}$, respectively. Additionally, the top drag coefficient is $C_d = 2.5 \times 10^{-3}$. The conservative temperature, absolute salinity, and velocity are averaged over the top mixed layer, defined here as a 20-m layer at the top of the cavity (or the entire top level where top levels are thicker than 20 m). The ice shelf thickness is static, so it is assumed that the ice dynamics instantaneously compensate melt-induced thinning.

The modeling domain is on a beta-plane with 64 regularly spaced points in both x and y, spanning ~502 km (horizontal resolution of ~8 km). The ocean floor is limited to 2000 m and is represented by 45 evenly spaced vertical levels. Walls exist on all four boundaries where the only external forcing is a restoring condition at the northern boundary, where the restoring is towards the initial state. The simulations are initialised from rest with initial conditions taken from the synthetic temperature and salinity fields described in Sect. 2.3.2, these fields then also set the northern restoring condition as simulation evolves. The configuration has no: surface forcing, sea ice and tides, but is inspired by the idealised ISOMIP+ experiments of Asay-Davis et al. (2016), where the interest here was to have a simple system in which to test the capabilities of a neural network to predict melt rates within drastically idealised ice shelf cavities. Future work will look at extending the neural network to more complex systems. Following Holland et al. (2008), all simulations in this paper have a common spin-up of ten years where the time-mean values of the final year are used for all analysis. Sensitivity tests (not shown) suggest that a 10 year spin up is sufficient to capture the equilibrated response of the ice shelf to the forcing.

## 2.2 Deep learning methodology

Our deep learning approach consists of two separate neural networks, trained to perform the two steps required to go from input fields to a melt rate field. All network design and training was done using MATLAB's deep learning toolbox (The MathWorks, 2021). In the first step, input geometries and ocean conditions (Sec. 2.3), together with NEMO melt rates (Sec. 2.1), are used to train a segmentation network that learns to classify regions of an ice shelf with labels representing the magnitude of melting or refreezing. Secondly, a denoising autoencoder network is trained to convert from these discrete labelled melt rates to a continuous melt rate field. Hereafter, we refer to the combination of these two networks working in tandem, which together form our proposed melt rate parameterisation, as MELTNET. More discussion on the use of two networks and comparison to more typical architectures can be found in Sec. 4 and App. E. Figure 1 shows the workflow for training each network and predicting melt rates and each of these steps is described in more detail below.

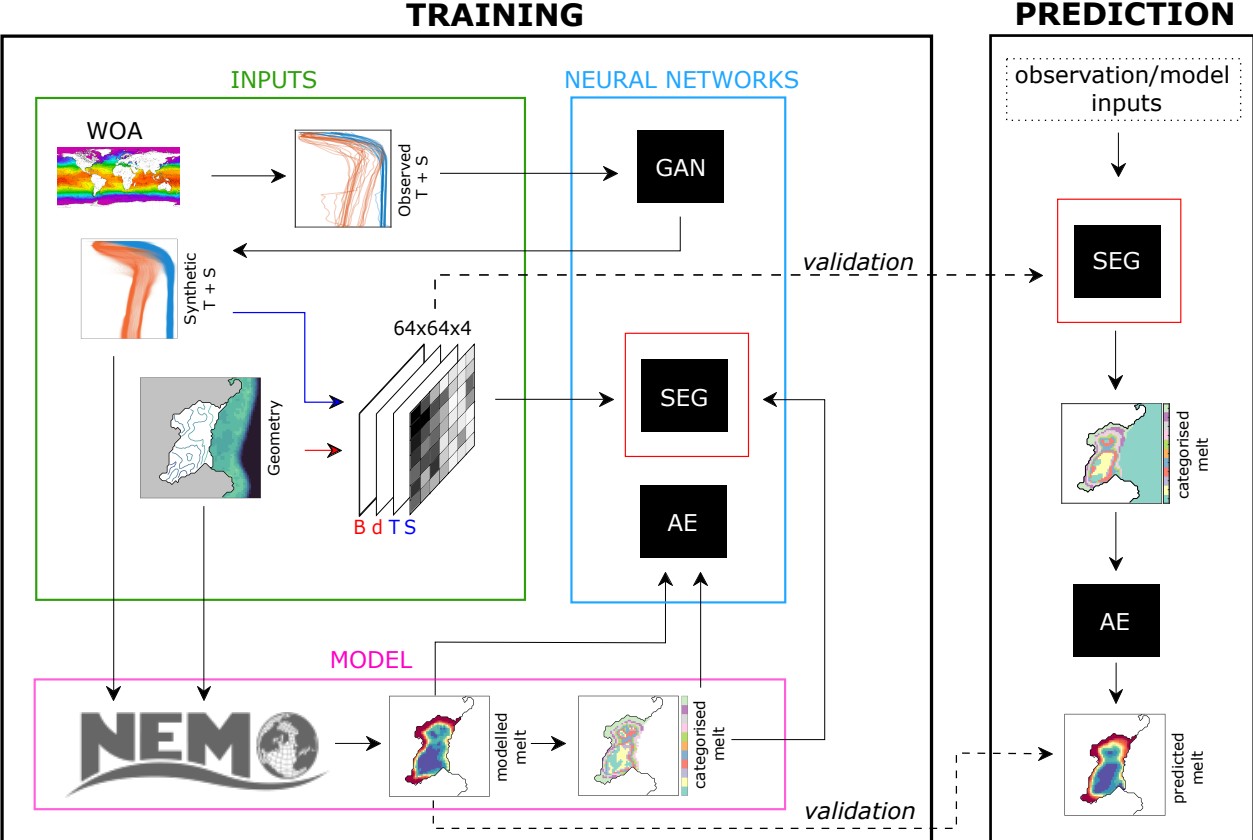

**Figure 1.** Workflow diagram for the proposed deep learning framework, split into training and prediction. Synthetic ice shelf geometries (Section 2.3.1) and synthetic temperature and salinity profiles generated from WOA data (Section 2.3.2) are used (1) as inputs for the NEMO model which predicts a melt rate field and (2) to create a four channel input image for training of the segmentation network. NEMO melt rates are converted into a labelled image and the segmentation network trains to segment input images that match labelled NEMO output. Separately, the autoencoder network takes the melt rate map and labelled melt rates from NEMO and learns to map between the two. In both of these networks, ~18% of inputs and NEMO melt rates are withheld to form the test and validation sets. Once the networks are trained, melt rate prediction proceeds by passing input images to the segmentation net and the resulting labelled images to the autoencoder, leading to a melt rate field. Note that the GAN is only used for generation of input fields, and not needed for melt rate prediction, as described in Sec. 2.3 and Appendix D.

### 2.2.1 Segmentation

The primary network, designed to classify melt rates from an input image, is a modified version of the U-Net architecture, originally proposed by Ronneberger et al. 2015 and harnessing subsequent extensions by Jha et al. (2019). A segmentation

network takes images as input and assigns a label or category to each pixel of that image. That input image may have different numbers of bands, for example a black and white image would have one input band and a standard colour image would have three. In our case, we use input images with $64 \times 64$ pixels and four bands representing bathymetry, ice shelf draft, temperature and salinity and output $N$ melt rate categories representing different regimes e.g. strong melting or weak refreezing. Note that the methodology is completely flexible with regards to the size of the image and the number of bands so more information could be coded into additional bands, as discussed later. In MELTNET, each pixel may have a value from 0 to 255 and the mapping from synthetic input fields to pixel values is explained in Sec. 2.3.

The target melt rate field output by NEMO (Sect. 2.1) must be converted to a labelled image with $N$ classes in order to be used to train the segmentation network. A tradeoff exists when selecting the number of classes for the segmentation network and the final performance of MELTNET in terms of predicting melt rates. With fewer classes, the segmentation net accuracy goes up but the inverse classification net struggles to infer complex melt rate patterns, whereas with more classes the segmentation net accuracy drops, also resulting in a drop in overall MELTNET performance. We tested networks using $N = 5$ to 12 classes and the resulting NRMSE (Normalised Root Mean Squared Error, described later) varied from 0.15 to 0.11. Based on this testing, an optimal number of classes for our training set was found to be $N = 10$. Melting (or freezing) rates were converted to $N$ discrete melt labels by calculating $N - 2$ quantiles of melt rates for every pixel in the training set and assigning labels to melt rates that fall between each quantile, with the last label reserved for regions of the image with no melt/refreezing (i.e. outside of the ice shelf).

The segmentation network takes these input images and the corresponding set of target melt rate labels from NEMO and learns to reproduce the labelled melt rate distribution. At its core are convolutions, consisting of sets of filters that operate on the image. The weights that make up these filters are learned during the training, by calculating their gradients with respect to a cost function and updating them iteratively to reduce the misfit to NEMO model targets. Layers of filters learn to extract useful features at different scales within the image, for example the outline of the coast or the local slope of the ice shelf base. The final training set, once a small subset of anomolous NEMO simulations with extreme temperatures were removed, consisted of ∼9,000 images, with a further ∼2,000 retained for validation and testing. MELTNET accuracy increases with an increasingly large number of training images but by testing incrementally larger training sets this number was found to be sufficient (Fig. B2). More details on the architecture, loss function and training of the segmentation net can be found in Appendix B.

### 2.2.2 Inverse classification

We use another deep learning approach to perform the task of converting from discrete melt labels, output by the SegNet, to a continuous melt rate field. We found that a modified denoising autoencoder (DAE) architecture, based on the network proposed by Zhang et al. (2017), was able to perform this task effectively. DAEs take partially corrupted input and are trained to extract features that capture useful structure in order to recover the uncorrupted original. In this case, the corruption is the process of categorising melt rates into $N$ discrete labels, which results in images that retain much of the original melt rate pattern but lose fine-scale detail and magnitude information. The segmentation net is trained on these labelled melt rate images rather than the

NEMO output directly, and itself outputs the same labels which need to be converted back to a continuous melt rate field in order to provide useful output for an ice sheet model.

The training set consists of labelled NEMO melt rates as inputs and true NEMO rates as outputs, i.e., the DAE learns to map from discrete labels to a continuous melt rate field. The specific experiments that comprised training, validation and test sets were the same as those used to train the segmentation network. This ensured that the DAE did not get any unfair advantage from having already seen similar melt rate patterns during its training as those output by the segmentation net. The DAE architecture consists of several layers of 2D convolutions, batch normalisations and Swish layers, as shown in Fig. C1 and described further in Appendix C.

## 2.3 Synthetic input generation

A major hurdle to overcome with this deep learning approach is to generate synthetic inputs that are realistic but also show sufficient variability to be useful analogs to real ice shelves. This problem can be broken down into two main steps; generation of the ice shelf geometry and generation of the temperature and salinity fields which set both the ocean initial conditions and far field restoring. Through the procedure described below we generate ~11,000 synthetic inputs. These are divided into a training set (~9,000), a validation set (~1,000) and a test set (~1,000).

### 2.3.1 Ice shelf and coastline geometry

Two of the four input bands that serve as inputs for the segmentation network consist of (1) ice shelf draft (i.e. ice thickness below water level defined within the ice shelf extent) and (2) ocean bathymetry, both beneath the ice shelf and in the open ocean. Generating these fields begins by defining three polylines: a randomly generated coastline, the ice shelf front position and the continental shelf boundary. Along the ice shelf boundary a grounding line thickness is defined, with deeper grounding lines far from the ocean and in narrow embayments to mimic ice stream in-flow to the ice shelf. Ice shelf thickness is advected from the grounding line to the coast using an analytical solution and finally a randomly generated bathymetry is defined based on the continental shelf boundary while ensuring that a cavity persists beneath the ice shelf. These steps are described in more detail in Appendix A. The result of this algorithm is two 64x64 arrays of ice shelf draft and bathymetry which are linearly re-scaled to pixel values from 0-255 in the first two bands of the segmentation input.

A sample of 36 synthetic domain geometries is shown in (Fig 2). The algorithm that generates synthetic ice shelf geometries must be capable of creating a wide variety of configurations. Validating these geometries is not possible, however the resulting configurations are visually similar to ice shelves typically found around Antarctica and the generation of ice thicknesses for each geometry, which melt rates are highly sensitive to, is based on analytical solutions for ice shelf flow. The final 64x64 grids result in each domain having an area of ~252,000km. Given that much of the domain is taken up by grounded ice/ocean, this results in maximum ice shelf areas which are less than the two largest ice shelves in Antarctica (the Ross and Filchner-Ronne Ice Shelves) but comparable to the next largest, such as the Amery and Larsen Ice Shelves.

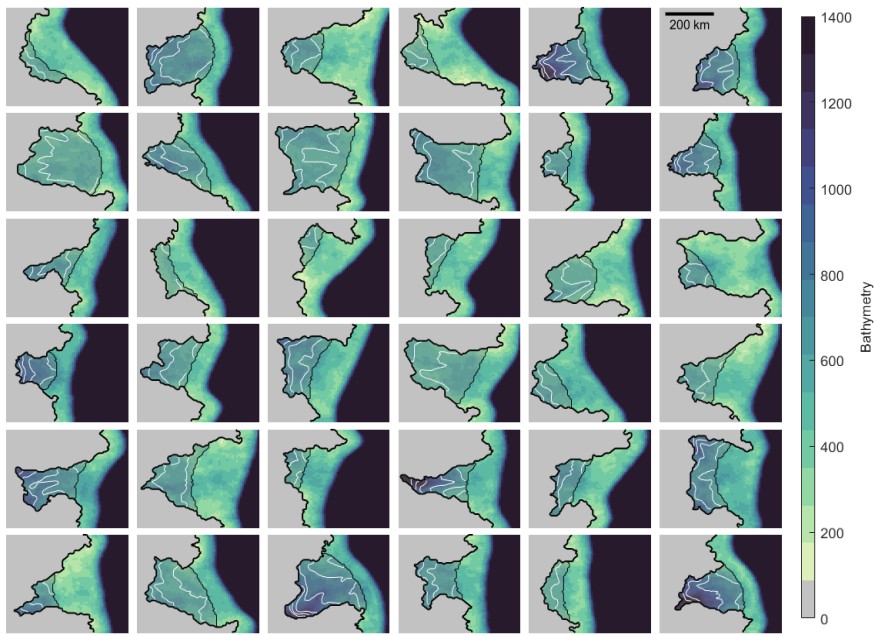

**Figure 2.** Sample of synthetic geometries produced using the algorithm outlined in Section 2.3.1. Grey delineates grounded ice and the colourmap represents seafloor bathymetry. White lines show ice shelf draft, contoured from 200 to 800 m at 100 intervals, and black lines show the outline of the ice sheet and ice shelf regions.

### 2.3.2 Temperature and salinity forcing

The geometrical inputs described above, that we vary spatially but not temporally, can be directly used as inputs to both MELTNET and NEMO since they both operate within the same 2HD space. In contrast, NEMO only uses temperature and salinity forcing at its boundary and as initial conditions, allowing these fields to evolve within the domain with time and contributing to the spatially varying melt rate field. Thus, temperature and salinity $T/S$ inputs for the two models need to be treated slightly differently. We first address how a forcing is generated for the ocean model, and then explain how this boundary forcing is mapped to the 2D inputs used by the segmentation network.

The ocean model configuration used here (Sec.2.1) requires a temperature/salinity ($T/S$) restoring condition designed to imitate the far-field ocean forcing of an ice shelf. In this case, the restoring condition is applied only at the northern boundary, similar to the ISOMIP+ experiments (Asay-Davis et al., 2016). We thus need $T/S$ fields that represent the variety of conditions that might be found in this location around Antarctica. To this end, we use the World Ocean Atlas 2018 (hereafter WOA; Boyer et al. 2018) to extract $T/S$ profiles around all of Antarctica. For each meridian in the gridded data, we take data from the first ocean grid cell north of the Antarctic coast that has a depth of more than 2000 m. While this results in several hundred vertical profiles that could be used directly to force our synthetic geometries, the WOA dataset is still inherently a finite source of $T/S$

data. As an alternative, we use these data as a starting point to generate synthetic $T/S$ profiles that share the same characteristics but can be unlimited in number and variety. This is accomplished with a Generative Adversarial Network (GAN, Goodfellow et al. (2014)). Details on this GAN network, together with a comparison between observed and generated $T/S$ profiles, can be found in Appendix D.

The depth-varying $T/S$ fields, generated by a GAN used as inputs to NEMO as described above, need to be mapped to the same 64x64 grid as the geometry for input to MELTNET. We explored several methods, most notably (1) mapping $T/S$ profiles that NEMO is forced (prescribed) with at the boundary directly to the ice shelf base at equivalent depth and (2) using the average $T/S$ profile, as simulated by NEMO at the ice front, and mapping this to the ice shelf base in the same way as (1). In practice, we found that it made little difference to the segmentation network accuracy (92.6% classification accuracy vs. 93.4%, respectively). We decided that taking average temperature and salinity conditions at the ice shelf front after model spinup was most consistent with existing melt rate parameterisations, since this is akin to forcing our model with observations near the relevant ice shelf. Furthermore, we view this as more useful to ice sheet modellers who might use MELTNET for future projections since it is generally believed that changes at the ice front (e.g. thermocline depth) are most closely related to the future evolution of Antarctica. These mapped $T/S$ fields are then linearly re-scaled to pixel values of 0-255 and form the remaining bands two of our input image.

## 2.4 Alternative melt rate parameterisations: PICO and PLUME

The performance of MELTNET is compared to two intermediate complexity melt rate parameterisations, where all models are judged on their ability to match NEMO's melt rate fields. The parameterisations are the Potsdam Ice-shelf Cavity mOdel (PICO, Reese et al. 2018a) and a 2D implementation of the plume model (Jenkins, 1991) based on Lazeroms et al. (2018) (referred to hereafter as PLUME). The PICO model includes a representation of the vertical overturning circulation within an ice shelf cavity with a series of boxes that transfer heat and salt from the grounding line to the ice front. The PLUME model adapts 1-D plume theory by selecting a melt plume origin at any given ice shelf point and determining melt rate as a function of properties at this plume origin and local ice shelf conditions. Plume origin for every ice shelf point is selected as the closest grounding line point, scaled by grounding line depth so that deeper origin points are favoured. Many other melt rate parameterisations exist but these were selected since they are generally regarded as the more advanced parameterisations; including physics related to cavity circulation while still remaining computationally inexpensive (Favier et al., 2019; Jourdain et al., 2020). For both parameterisations, a high resolution version of the synthetic geometries was converted to a finite element mesh and the Úa ice-flow model implementation of each model was used to calculate melt rates.

In order to make our comparison to the PLUME and PICO models as fair as possible, two uncertain parameters in each model were optimised using the training set of NEMO outputs. For the PICO model, these two parameters were the overturning strength ($C$) and the heat exchange coefficient ($\gamma_T^*$), which are also treated as tuneable parameters in the original PICO paper (Reese et al., 2018a). For the PLUME model, the heat exchange parameter $\Gamma_{TS}$ (similar but not the same as the $\gamma_T^*$ parameter for PICO) was selected, together with the plume entrainment coefficient ($E_0$). The PLUME and PICO models were run using the input geometry, temperature and salinity fields and then a cost function as calculated as the total Normalised Root

Mean Squared Error (NRMSE) compared to NEMO melt rates. This cost function was then minimised using the constrained optimisation function 'fmincon' in MATLAB, to derive an optimal set of parameters for each model that most closely replicated the NEMO melt rates. The optimised values of all intermediate complexity model parameters, together with the values originally suggested in their corresponding papers, are shown in Table F1.

## 3 Results

Figure 3 presents the main result of our study, a grid of different geometries (row 1) and corresponding melt rates ($m\,yr^{-1}$) as calculated by: NEMO (row 2), MELTNET (row 3), PICO (row 4) and PLUME (row 5). Melt rates calculated by MELTNET clearly stand out amongst the lower three panels as the best qualitative match to the NEMO ocean model. In order to represent the range of performance, rather than just the best results for any particular parameterisation, we assign a score to each MELT-NET result and sample evenly from the distribution of results by calculating the quantiles of the scores. The score used here (row 6), is based on a combination of the NRMSE and correlation coefficient; the two quantities are treated as vectors whose length is scaled such that a vector of zero length is a perfect score (NRMSE of 0 or correlation coefficient of 1) and the score is the L2 norm of these two vectors. Row 6 of Fig. 3 therefore shows the scores for all experiments from the test set ($N = 951$) in blue and the specific experiments sampled from the distribution of these scores (as plotted in the rows above in the same figure) are marked in red. Panels for individual experiments are sorted by increasing score from left to right. In the bottom left corner of each panel we show the averaged melt rate as calculated by each model. Only geometries from the test set, i.e. geometries that MELTNET did not see during training, are included in this comparison.

Model performance as measured by NRMSE and correlation coefficient is shown for the three models and for all members of the test set in Fig. 4. Model misfit, in terms of NRMSE, was lowest with MELTNET for 96% of the members of the test set (panel a). Pearson correlation coefficient between MELTNET and the NEMO model (mean 0.73) was higher than PICO (mean 0.25) and PLUME (0.14) for 99% of the members of the test set (panel b). These results show that MELTNET not only has a lower misfit than the other models but is far better at reproducing spatial patterns i.e. getting melting and refreezing in the right areas of each ice shelf.

### 3.1 Idealised Experiments

A commonly raised and sensible concern with all forms of deep learning is that poorly trained networks can give the right answer for the wrong reasons. We take a number of steps to avoid this issue, for example, we randomly rotate the ice shelf orientation so that the network does not learn to associate high melting with grid cells on one side of the domain. We can also explore how MELTNET's predictions compare to our understanding of the physics underlying ocean-ice shelf interactions. This is done for a simplified ice shelf geometry, i.e., uniform bathymetry at a depth of 1600m and an ice shelf 120 km long and 280 km wide. Ice shelf draft varies linearly from 600m at the grounding line to 200m at the ice front and there is no across-shelf variation in the geometry. Furthermore, both salinity and temperature are constant throughout the entire domain, and as before, this forms the northern boundary restoring condition. In this context, two simple relationships are expected to

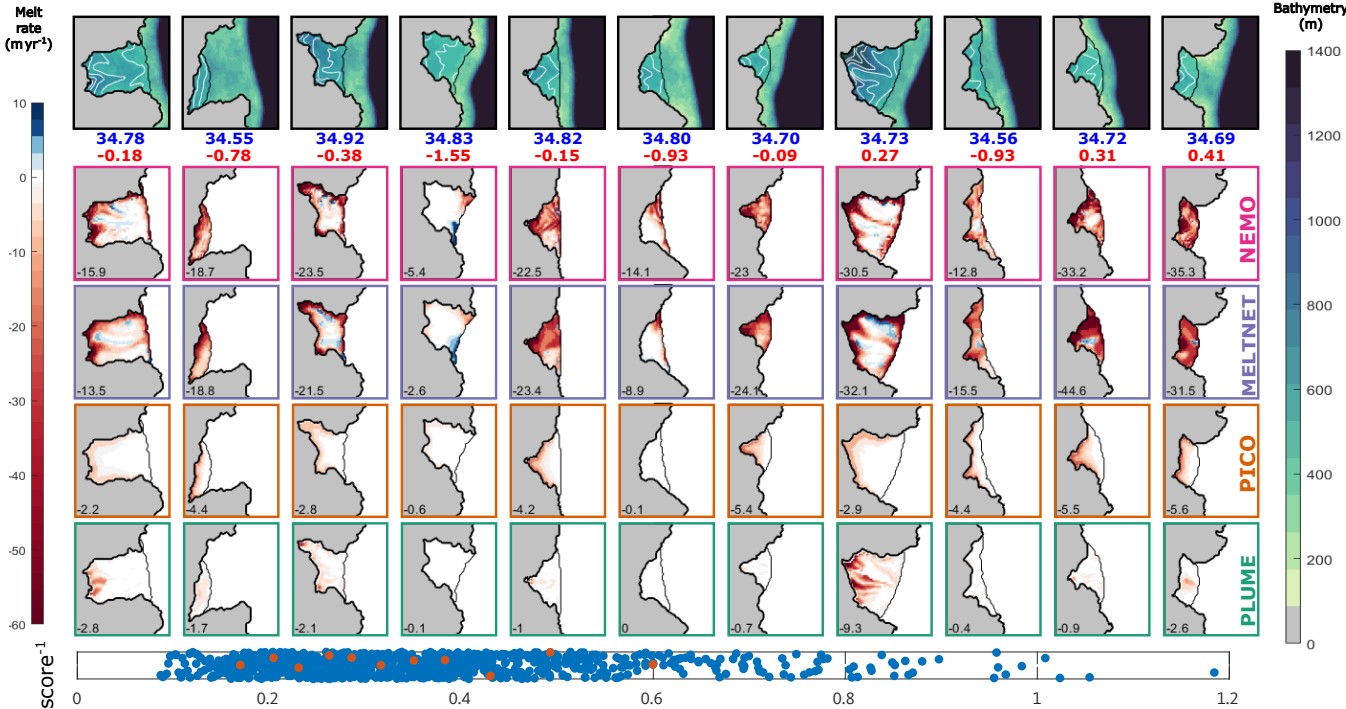

**Figure 3.** Input geometry (row 1) for a sample of synthetic ice shelves and the resulting melt rates (m yr$^{-1}$) as calculated by NEMO (row 2), MELTNET (row 3), PICO (row 4) and PLUME (row 5). Experiments were selected by sampling evenly from the distribution of MELTNET scores. Scores, calculated as a combination of NRMSE and correlation coefficient, are shown in blue in row 6 and the sampled experiments are highlighted in red). Only geometries from the test set are included in this figure. Background colourmap for the geometries shows ocean bathymetry (scale on the right hand side) and white lines show ice shelf draft, contoured from 200 to 800 m at 100 m intervals. Melt rate results all use the same colourmap, with red and blue indicating melting and refreezing, respectively. Note the colour map gradient is not linear, but is greatest around zero, to make it easier to distinguish the magnitude of melting/refreezing over the bulk of the ice shelves. Numbers in red and blue at the top of each melt rate column show the area averaged sub-ice-shelf temperature and salinity, respectively. Numbers in the bottom left corner of each melt rate panel show the averaged melt rate as calculated by each model. Both the PICO and PLUME models show results using optimised parameters (Table F1).

emerge: (1) a linear dependency of melt rates to changes in ice-shelf slope (Jenkins, 2018) and (2) a quadratic dependency of melt rates to changes in ocean temperature (Holland et al., 2008). To investigate these two relationships, we: (1) vary ice shelf
basal slope by keeping grounding line depth constant and moving the ice front and (2) vary temperature by a uniform amount in the entire domain. The change in the ice-shelf cavity melt rate for these two sensitivity tests is shown in Fig 5. In both tests, the dependence matched that expected by theory, as shown by the linear and quadratic trend lines through the sample points. This goes some way to demonstrating that MELTNET has learnt an accurate representation of actual melt physics. This is in spite of the fact that these simplified domains are very different from the more complex geometries that the network has been
trained on.

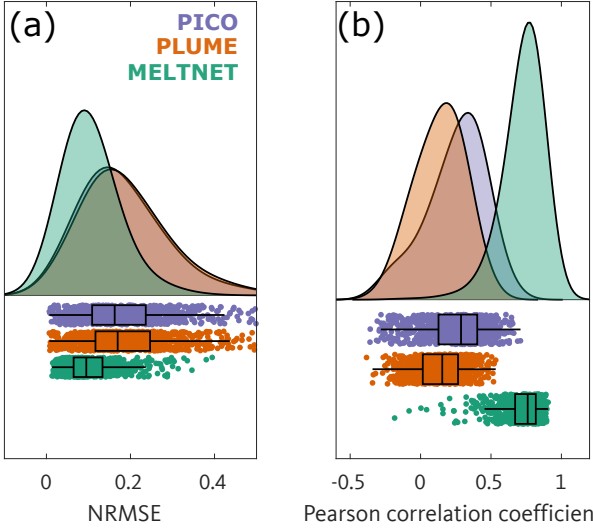

**Figure 4.** Distributions of NRMSE (panel a) and Pearson correlation coefficient (panel b) for the three different melt rate parameterisations: PICO (violet), PLUME (orange) and MELTNET (green). The lower section of each panel shows each individual score from the entire test set, overlain with a boxplot. The box represents the interquartile range of scores and the vertical line through the box is the median. The upper section of each panel presents the same information as probability density plots. Note lower NRMSE and higher correlation coefficient mean a better fit to the groundtruth NEMO melt rates.

## 4  Discussion and Conclusions

The MELTNET deep neural network can produce melt rates that closely resemble those calculated by the NEMO ocean model for synthetic geometries that were not part of the training set. When compared to two intermediate complexity melt rate parameterisations, MELTNET outperforms them in terms of both overall NRMSE and correlation, even when parameters in those models are tuned to minimise the misfit for the geometries we test. In terms of area averaged melt rates (Fig 3), MELTNET also performs favorably compared to PICO and PLUME, which both tend to underestimate this value. Since these two models are tuned to minimise the overall NRMSE rather than average melt this is not particularly surprising, but nevertheless highlights the problem with tuning these models based on one metric, leading them to perform poorly in other regards.

Correctly predicting spatial patterns of ice shelf melting (as shown by the high correlation of our results to NEMO), rather than just the magnitude, is crucial because the sensitivity of an ice shelf to thinning will vary across that ice shelf (Reese et al., 2018b). Some regions within an ice shelf can be considered entirely 'passive', in that reductions in ice thickness in these areas has no impact on ice flux across the grounding line. On the other hand, perturbations to ice shelf melting within certain highly buttressed regions, for example in the shear margin of an ice shelf, can have a much greater impact on ice-sheet discharge than other regions such as downstream of an ice stream, despite those being otherwise dynamically important (Feldmann et al.,

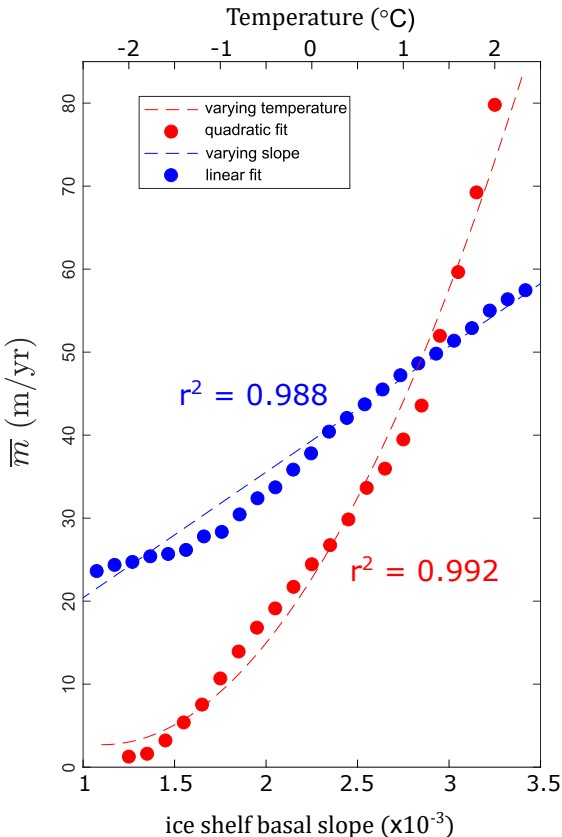

**Figure 5.** Area averaged basal melt rate from MELTNET for an idealised ice shelf geometry as a function of changes in ocean temperature (red) and ice shelf basal slope (blue). Dashed lines show quadratic and linear fits to results, with corresponding $r^2$ values in matching colours.

2021). In general, a small proportion of ice shelves contribute a disproportionately large fraction of the total buttressing force (Reese et al., 2018b).

A clear motivation behind using a deep learning approach is that the computational cost is almost entirely associated with training the network (i.e. everything contained within the training box of Fig. 1), whereas predicting melt rates using the trained network is extremely fast. Producing the training set by running NEMO on our synthetic geometries required significant computational resources, although since the problem is embarrassingly parallel this is not necessarily a major hurdle. Running the $\sim$11,000 NEMO simulations was done on the ARCHER2 Cray EX supercomputing system which has 5,860 compute nodes, each with 64 dual core CPUs. Each simulation used 20 cores and the total compute time for all NEMO simulations was $\sim$1.3 million core-hours. Training the two networks was done using a single NVIDIA K5200 GPU and took $\sim$48 hours for

the segmentation network and ∼6 hours for the DAE network. Once trained, MELTNET can predict melt rates from inputs in 20 ms. A direct comparison to the average time taken by NEMO to calculate melt rates is difficult, particularly given that we spinup NEMO for 10 years prior to making use of its melt rate predictions. In a coupled transient configuration the speedup would depend strongly on the coupling timestep, since an ocean model invariably requires a far smaller timestep than an ice sheet model. Assuming a coupling timestep of 1 year and for this configuration of NEMO the speedup would be between 4-5 orders of magnitude. Compared to PICO and PLUME the speedup is approximately 1-2 orders of magnitude depending on the configuration of those models that we use.

One important caveat to this work is that MELTNET can only be, at best, as good as the ocean model that it has been trained on. Here, we necessarily treat the ocean model as our ground-truth, since the geometries are entirely synthetic. Training MELTNET on real world observations would be preferable but there are not enough distinct ice shelves, or indeed sufficient observations of melting, for this to be feasible. Thus we consider NEMO melt rates akin to observations and matching these as accurately as possible is our goal. The NEMO ocean model setup has a number of simplifications; for example no representation of sea ice, surface forcing, ocean tides etc. These processes would all impact the melt rate calculation. Some missing processes would be possible to add in with our synthetic geometry approach, but others present more significant challenges.

On the other hand, this methodology also provides interesting advantages, since adding complexity to the representation of the ocean model physics can simply be achieved by including more processes in the ocean model. This is in contrast to a typical model where adding new physics is a significant undertaking that can require replacing large sections of code and considerable testing. Furthermore, since the method is not limited in terms of input fields, any missing information required to properly train the network with new physics could easily be added into a new band in the input image. Conversely, processes could be removed by reducing the amount of inputs used to train MELTNET. Doing this would provide insights into which processes are important for producing realistic melt rates, possibly aiding the development of alternative parameterisations. This also alludes to a unique advantage of a deep learning emulator such as MELTNET: its success in retaining the important physics of complex models in a highly efficient way. Intermediate complexity models such as PICO and PLUME use simplified systems of equations, based on our understanding of a physical system, but require a small number of parameters to be tuned in order to match observations. With a deep learning approach the degrees of freedom are many orders of magnitude higher, so in a sense it is infinitely tunable, but through training the network learns which relationships between input and melt rate are relevant without needing to simplify or discard physics that may be important. For example, neither PICO nor PLUME currently account for the coriolis effect, which is known to affect the melt rate distribution on an ice shelf through its role in determining the direction of currents beneath the shelf. MELTNET, however, implicitly includes the coriolis effect in its melt rate calculations, since this contributes to the melt rates calculated by NEMO and so its effects can be learned from the training set.

The MELTNET model presented here stacks two networks, the segmentation and DAE components; trained separately and with very different architectures. These two networks split the task of predicting melt rates for an ice shelf into two sub-tasks. The segmentation network converts a multi-channel input image into a segmented single band image in which regions of varying melt are labelled and the DAE converts labelled melt rates a melt rate prediction. The task performed by MELTNET

can also be attempted by a single neural network, for example a convolutional neural network (CNN) with a regression output layer (e.g. Jouvet et al. 2022 is a recent application of this in the field of glaciology). Alternatively, the segmentation network can be easily converted to directly predict a continuous melt rate field, rather than melt rate labels, by changing the output layer and loss function. We explored these options and others during the development of MELTNET, but these were all considerably outperformed by MELTNET. The use of two networks to solve this type of problem has not been done before to our knowledge and undoubtedly will not be suitable in many other cases. We discuss this in more detail and present some results from other architecture choices in Appendix E.

The results presented here show a promising first step to a parameterisation for ocean induced melting that shows high fidelity to advanced ocean models with very low computational cost. That being said, more work is required, before applying this to transient ice sheet models. Future work must demonstrate that the network, trained on synthetic geometries, is also capable of reproducing melt rate patterns on real ice shelves based on the limited observations that exist or in comparison to state-of-the-art ocean models. One limiting factor currently is the size of the domain, which at $\sim 502\,\mathrm{km}^2$ is not large enough to cover the very largest ice shelves and include neighbouring open ocean to serve as input for training and prediction. Furthermore, more work is needed to show the equivalence of these synthetic geometries to real ice shelves, include complex bathymetric features such as ice rises or troughs that play an important role in providing access for warm water to the grounding line, and add more processes to the ocean model that is used for training. Subsequently, MELTNET would need extensive testing to ensure that fringe ice shelf configurations never lead to extreme melt rate predictions that are outside of what would be physically plausible. A recent innovation that could help in this regard is the development of neural network emulators that are constrained, either through the loss function or their architecture to ensure that they do not violate physical laws (e.g. conservation) (Beucler et al., 2021).

This work has demonstrated that a deep learning network can be trained to emulate an ocean model in terms of predicted melt rates beneath an Antarctic ice shelf. When applied to a wide range of synthetic geometries, MELTNET agrees closely with the NEMO model that it was trained on, and outperforms other commonly used parameterisations if we assume that the ocean model represents the best estimate of melt rates for a given geometry. These results show that a deep learning emulator may provide useful melt rate estimates for ice sheet models but more work is needed to refine the methodology and test this approach on real ice shelf geometries with observations of melt rates. An accurate and efficient parameterisation of melt rates beneath Antarctic ice shelves is urgently needed to improve the representation of this crucial component of mass loss.

*Code availability.* All deep learning development and training was done using MATLAB's deep learning toolbox (The MathWorks, 2021). MATLAB code for training of and prediction with MELTNET, including a sample of the validation sets, is available at https://zenodo.org/record/7018247 (Rosier, 2022). The two intermediate complexity melt-rate parameterisations were implemented in the open-source ice flow model Úa, available at https://doi.org/10.5281/zenodo.3706623. A complete set of the synthetic geometries, generated to train MELTNET, is available from the authors upon request.

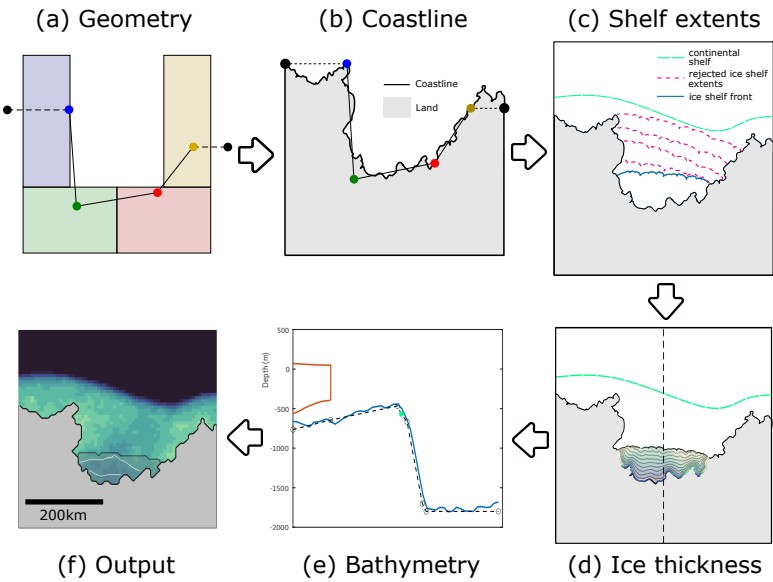

**Figure A1.** Flowchart showing the steps leading to generation of a random ice shelf geometry, consisting of (a) generation of corner nodes randomly located within the four coloured polygons, (b) creation of a continuous fractal coastline, (c) definition of ice shelf and continental shelf horizontal extents, (d) definition of ice thickness at the grounding line and from this the full 2D ice shelf draft, (e) generation of a bathymetry constrained by the previous steps, leading to (f) one randomly sampled ice shelf geometry (background colourmap represents ocean bathymetry and contours are ice shelf draft, following the same ranges as in Fig. 3).

## Appendix A: Synthetic Ice Shelf Generation

Here, we describe in more detail the process of generating synthetic ice shelf geometries, which form two inputs bands of the training images as explained in Sec. 2.3. The creation of the bathymetry (including the coastline) and ice-shelf draft can be broken down into four steps. The algorithm consists of:

1. Create an outline of the coast.

2. Define the ice-shelf and continental shelf horizontal extent.

3. Define ice thickness at the grounding line and then generate ice shelf draft.

4. Define bathymetry, constrained by features generated in preceding steps.

Each of these steps generates one or more random numbers that determine some geometrical property of the final domain, leading to a very large variety of final ice shelf configurations. These steps will now be outlined in order, with each step in its
own paragraph.

The starting point for generating ice shelf geometries is the observation that most Antarctic ice shelves, particularly large ones, occur within embayments along the coast, while some smaller unconfined ice shelves also exist along flatter sections of

the coastline. From a square domain, we start by creating a polygon that will define the overall geometry of the coastline from four random point seeds that can each lie anywhere within their four predefined boxes, as shown in Fig A1a. Two further points are added on either side and inline with the previous end points to create a polygon with six points and five edges. Due to the extents of the four boxes within which the four initial points are seeded, most geometries will consist of a central embayment but the concavity of the resulting bay can vary from almost flat to a deep and strongly confined. This simple polygon is then transformed into a complex polygon more closely resembling the fractal nature of a real coastline by repeatedly adding points midway between two existing points and offset some random distance from that edge, resulting in a final coastline as shown in Fig A1b.

With a coastline defined as described above, the next step is to define plausible extent for the ice-shelf front and from this the continental shelf break. Points on the coastline nearest to the two corner points (blue and yellow points in Fig A1a) are selected as trial start and end points for the ice-shelf front which has a random curvature (this determines the ice-shelf front shape, from concave to convex). If the ice-shelf front polygon does not intersect any coastline points and the area is less than a randomly selected minimum area then the ice-shelf front is accepted. If the ice shelf front is rejected, the two starting points along the coast for the calving front polygon are moved closer together and the procedure is repeated until a geometry is accepted. This results in a variety of different sized ice shelves that tend to be confined by any existing embayment in the coast. As a next step, a distance is calculated for each open ocean point and the combined ice front and coastline ocean boundary. A contour of constant distance from the coastline is then drawn and converted to a tensioned spline to generate a smooth polygon that defines the continental shelf break. The result of these two steps is shown in Fig A1c.

The geometry is now fully defined in 2D, but requires information on ice and water column thickness to be used as input for the ocean model. Ice thickness is first defined everywhere along the ice shelf grounding line as a product between distance to the ice front, a measure of the coastline curvature and a random factor (resulting in a maximum ice thickness at the grounding line of 2000 m). This leads to ice thicknesses that are generally greater further from the coast and particularly where the coast consists of smaller inlets, to mimic plausible ice streams flowing into the ice shelf. Ice thickness is then extrapolated at regular points along the grounding line to the ice front using a simple analytical expression for a buttressed ice shelf thickness profile from Nilsson et al. (2017) under the simplifying assumption of no net accumulation. These ice thickness profiles are combined and mapped onto a grid to generate a 2D ice thickness field everywhere within the ice shelf (Fig A1d).

The ice thickness and continental shelf are used to constrain the ocean bathymetry by constructing polygons at regular intervals through the domain. Points at 2000 m depth are added at the open ocean boundary and a fixed distance from the shelf break. Further points are added around the shelf break at depths controlled by the ice shelf draft at the ice front and a final point is added at the grounding line at 200 m below the grounding line depth. The polygons are combined to form a grid of depths and random brownian noise is added to generate the final bathymetric grid (Fig A1e). The resulting fields of ice thickness and ocean depth are generated at finer resolution and then linearly interpolated onto 64x64 grids which serve directly as inputs to the ocean model (with each cell representing ∼8x8 km) and their discretised form serve as two bands of the input images for the segmentation net (Fig A1f).

## Appendix B: Segmentation network architecture and training

We use a modified U-Net architecture for melt rate segmentation, based largely on the recently proposed ResUNet++ architecture (Jha et al., 2019). The U-Net architecture, initially developed for medical image segmentation, has proven highly successful for a wide variety of segmentation tasks and spawned a number of U-Net-style architectures that add various blocks to improve performance. At its core, U-Net consists of a sequence of convolution blocks and pooling operations that reduce the spatial dimension of the image while increasing the number of feature channels (sometimes referred to as the encoder), connected via a bridge to upconvolution blocks (the decoder) that effectively operates in reverse resulting in a segmented image with the same dimensions as the input. Our implementation, adapted from ResUNet++, makes use of the following additional components:

1. Residual connections for each encoder and decoder block. These have been shown to improve training and propogation of information through deep neural networks (He et al., 2016).

2. Squeeze and excitation units preceding each encoder block, that adaptively weight feature maps to increase sensitivity to relevant features with very low additional computational cost (Hu et al., 2018).

3. Atrous Spatial Pyramid Pooling (ASPP), acting as a bridge between the encoder and decoder blocks and between the last decoder and the classification layer, that aid in capturing information at multiple scales (Chen et al., 2018).

We modify the ResUNet++ architecture by removing attention blocks and using Swish activation functions to improve performance. Unlike in classical medical image or driving segmentation tasks, where attention blocks help focus the network on relevant features in an image, we found this did not improve performance in our case, possibly because certain input layers are only non-zero within the ice shelf and so act as a shortcut for the network focus on that region. We found that using Swish activation functions, which have the form $f(x) = x \cdot \mathrm{sigmoid}(x)$, improved model convergence and alleviated an occasional issue of vanishing gradients. This finding is in line with other studies that have shown their advantage over ReLU activation functions (Ramachandran et al., 2017). The overall network architecture is shown in Fig. B1.

We trained the segmentation network on ∼9,000 training images using stochastic gradient descent with momentum (SGDM), an initial learning rate of $5 \times 10^{-7}$ and a momentum of 0.85. The SGDM algorithm updates network weights based on our loss function, for which we use a weighted cross entropy loss, which is a standard choice in classification tasks. Through this, we seek to minimise the difference between the predicted and target probability distributions of each class. We weight the loss function according to the normalized inverse frequency of labels in the training set, excluding the class representing 'no melt' by setting its weight to zero. Including this class in the loss function would result in a large bias in the training and scoring of the network towards pixels that are of no interest to our application. Weighting the remaining classes based on frequency helps ensure the network is equally successful at predicting less frequent classes, e.g. refreezing. The model was trained for 600 epochs with a small mini-batch size of 16, which tends to improve the models ability to generalise. Model hyperparameters including learning rate, batch size and number of filters were manually optimised, although due to the computational cost of training the network this parameter search was by no means exhaustive. In Fig. B2 we plot how the segmentation network

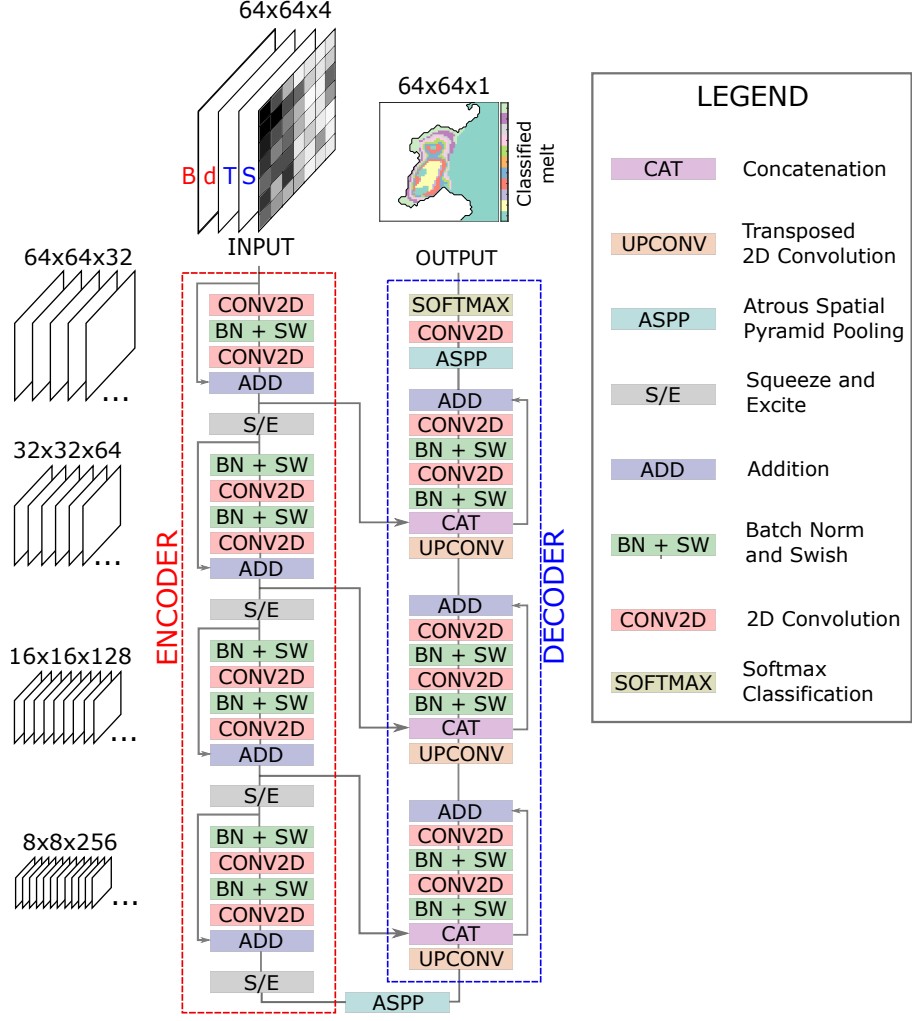

**Figure B1.** Schematic for the segmentation network used in this study. An input 64x64x4 image goes through layers of 2D convolutions, batch normalisations and swish layers through the decoding and encoding branches. Once trained on NEMO results, the final output is a segmented image consisting of melt rate labels that can then be converted into a melt rate field using the DAE network.

validation NRMSE and correlation coefficient changes with respect to the size of the training set, showing that the training set is sufficiently large for our purposes.

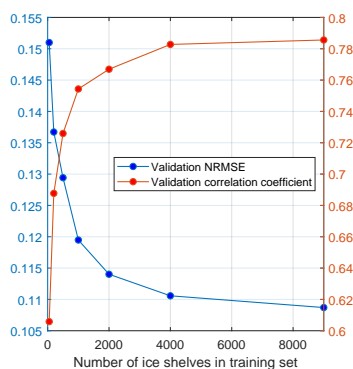

**Figure B2.** Change in NRMSE and Correlation Coefficient for MELTNET with respect to NEMO melt rates as a function of changing the size of the training set.

### Appendix C: Inverse classification network

The second component of MELTNET, designed to convert labelled melt classes to a continuous melt rate field and that we refer to as our inverse classification network, is based on a denoising autoencoder (DAE) architecture. This constitutes an input convolution layer, a series of four 2D convolutions each with 16 filters of size 3x3, activation functions and batch normalization layers, followed by a final convolution layer and a linear activation function, as shown in Fig. C1. Once again, for all but the last layer we use Swish activation functions rather than ReLU to improve convergence in training.

We train the network by minimizing the mean-squared error between the target and predicted melt rate fields using the SGDM algorithm with a learning rate of $2 \times 10^{-8}$, a momentum of 0.9 and a mini batch size of 16. We add regularization with an amplitude of $5 \times 10^{-2}$ that penalizes particlarly large weights, to avoid overfitting. The network is trained for 500 epochs on the same set of synthetic ice shelf experiments as the segmentation network (8922 images). Model hyperparameters, consisting

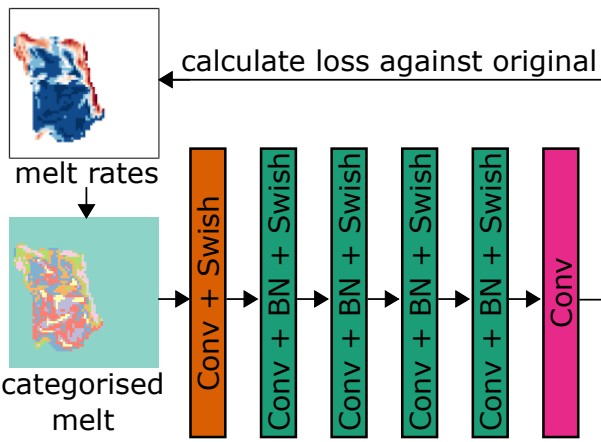

**Figure C1.** Schematic for the AE net architecture used in this study. A melt rate field, as calculated by NEMO for a given input geometry, is converted to a matrix of discrete melt labels. These labelled melt rates served as the input to the network go through a series of convolution, batch normalisation and swish layers, leading to an output. The loss of this output is calculated against the original melt rate field to train the network to recover a continuous melt rate field from discrete melt rate labels.

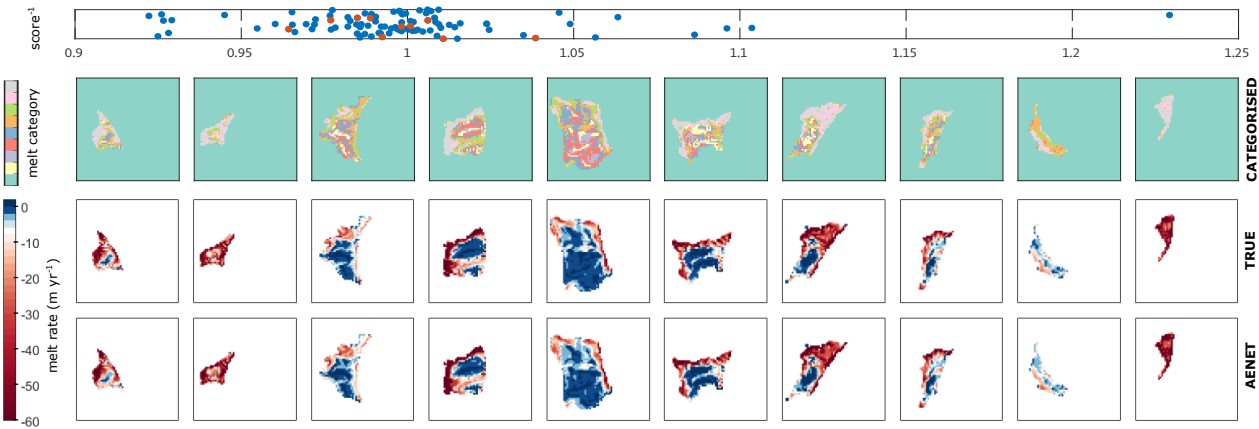

**Figure C2.** Example of the DAE network used to convert from labelled melt rates (top row) to a continuous melt rate field (true melt rates in the second row, predicted melt rates by the DAE network in the bottom row). Examples shown are selected by evenly sampling the distribution of DAE network scores (all scores shown in the upper scatter plot with sampled scores in red).

of the learning rate, momentum, batch size and regularization, were manually optimised. A comparison between NEMO melt rates, those same NEMO melt rates converted to a labelled image and the result of mapping from labelled images to melt rates using the trained DAE network is shown in Fig. C2.

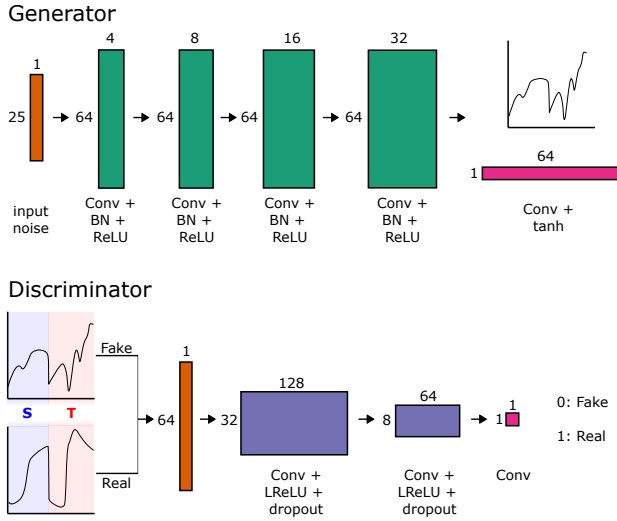

**Figure D1.** Schematic showing the GAN architecture, used to generate synthetic temperature and salinity profiles from the WOA observations. The Generator network (top) takes a 25x1 random vector and, through a series of convolution layers, outputs a synthetic combined profile. The discriminator network (bottom) is given both real and synthetic profiles and labels these as real or fake. Both networks learn from one another, resulting in a generator network that can create an infinite number of realistic temperature and salinity profiles from random noise.

## Appendix D:  Synthetic Temperature and Salinity generation

As described in the methods, rather than using the finite number of observations of temperature and salinity to force our synthetic geometries, we generate synthetic profiles using a GAN. The GAN consists of two networks trained together; a
generator network and a discriminator network. The generator learns to generate synthetic temperature and salinity profiles while the discriminator network attempts to distinguish between real profiles (from the WOA dataset) and the profiles created by the generator. Initially, neither network knows what to do but are in direct competition and learn from each other to improve. At the end of the training process, the generator network has learnt to take a random vector input as a seed and output temperature and salinity profiles that closely resemble the real data. Since the GAN takes a random seed as input, any number of these
random seeds can be used to generate the desired number of synthetic profiles.

The specific architecture used, shown in Fig. D1 is a modification of the Deep Convolutional GAN (DCGAN) as proposed by Radford et al. (2016). Temperature and salinity profiles from WOA are concatenated into one vector which the discriminator aims to reproduce and the discriminator learns to differentiate. The discriminator network includes dropout layers with a dropout of 50%, which was necessary to avoid mode collapse. The two networks are trained simultaneously for 500 epochs
and reach an equilibrium in which the loss for each stabilises around 0.5. Every available temperature and salinity profile from the WOA dataset, used to train the GAN, is shown in Fig. D2a, together with a sample of synthetic profiles generated by the GAN in Fig. D2b.

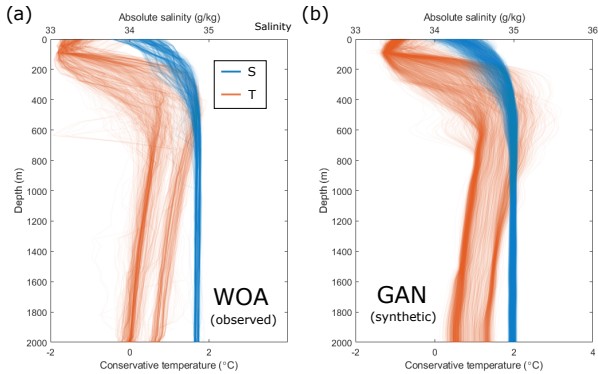

**Figure D2.** Temperature and salinity profiles (a) extracted from the WOA dataset (Boyer et al., 2018) from the closest 2000m depth cell around Antarctica ($N = 1440$) and (b) generated from the GAN ($N = 5000$), trained on the WOA data shown in (a). WOA data is converted to the TEOS10 standard (IOC and IAPSO, 2010), prior to being used to train the GAN.

## Appendix E:  Architecture choice

MELTNET consists of two networks acting in tandem: treating the task of melt rate prediction as a classification task and using a denoising autoencoder to convert this segmented prediction to a continuous melt rate field. This approach has not, to our knowledge, been used before to emulate physical models and yet, in this case, it shows improved performance compared to other architectures we tested that make melt rate predictions using a single network. A large number of alternative machine learning architectures could be used to solve the task of melt rate prediction directly, for example as a single network regression problem rather than the stacked network approach that we use in MELTNET, and we discuss some of these alternatives below.

A convolutional neural network (CNN) can be used to emulate complex physical systems, for example this methodology has shown great success in emulating glacier flow (Jouvet et al., 2022). We explored this option using a CNN consisting of a series of convolution operations followed by nonlinear activation functions and a final linear activation function in the output layer. The model was trained by minimizing a mean-squared-error loss function on the NEMO modelled melt rates and model hyperparameters (batch size, learning rate, filter size, number of layers and activation function choice) were tuned on the validation set to obtain the best results possible. Performance on the test set compared to MELTNET was considerably lower in terms of both NRMSE (0.134 vs. 0.105) and correlation (0.510 vs. 0.732).

Alternatively a UNET style architecture, such as the one used in the segmentation component of MELTNET, can be altered to predict the desired melt rates via regression, rather than melt rate labels. We explored various architectures here as well but we will focus our comparison on the simplest and fairest option: converting the segmentation network already used within MELTNET directly to predict a continuous melt rate field. This can be accomplished by changing the output layer to a linear activation function and minimising the same mean-squared-error loss function as the second network of MELTNET. This means our modified network has approximately the same number of learnable parameters as MELTNET (since the DAE component has relatively few). Model hyperparameters (including learning rate, batch size, momentum and filter size) were optimised

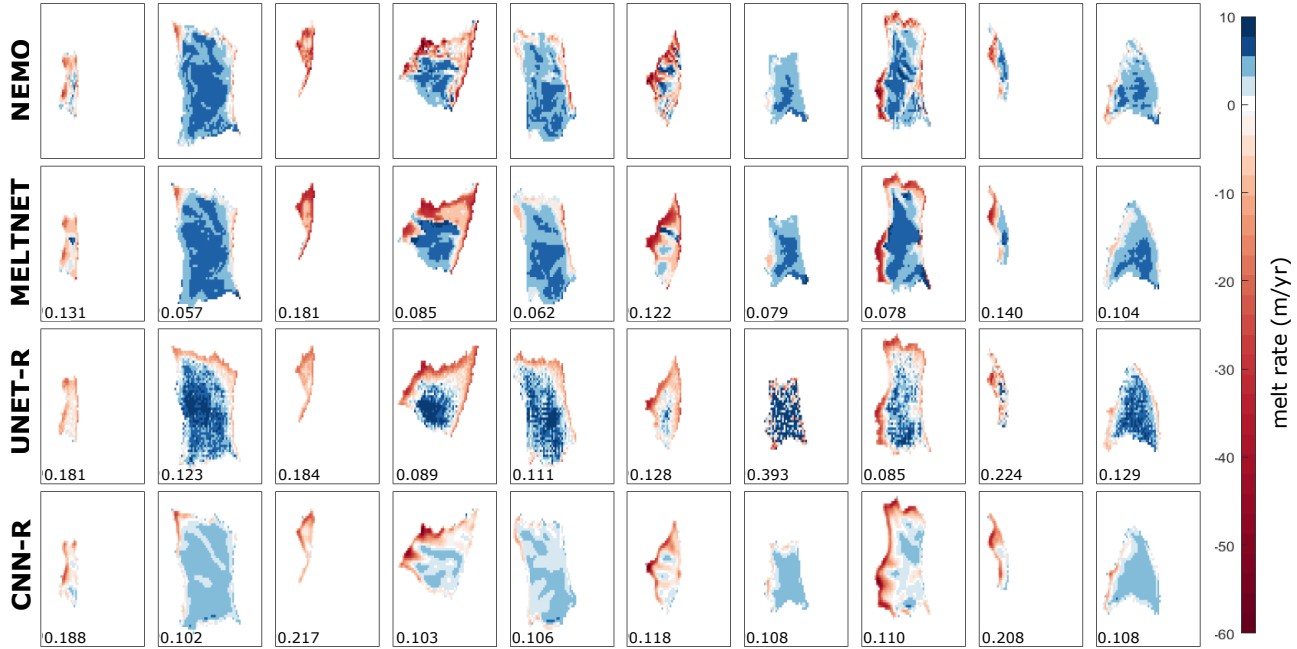

**Figure E1.** Comparison between a random selection of NEMO modelled melt rates (row 1), melt rates as predicted by MELTNET (row 2), melt rates predicted by an alternative UNET architecture with a regression output layer (UNET-R, row 3) and a regression CNN (CNN-R, row 4), as described in Sec. E. Numbers in the lower left corner of each neural network output show the NRMSE with respect to the NEMO melt rates.

using the validation set and performance on the test set was then compared to MELTNET. Once again, this architecture performs
considerably less well in terms of both NRMSE (0.135 vs. 0.105) and correlation (0.593 vs. 0.732). We show a comparison between melt rates calculated by NEMO, MELTNET and the two alternative architectures discussed in this appendix in Fig. E1.

Stacking two networks as we have done here in MELTNET represents a different approach to other emulators of physical systems using some form of convolutional network, which generally treat this as a regression problem. Here, instead, MELT-NET treats the prediction of melt rates as a segmentation task, seeking to identify which regions of an ice shelf will have a
certain melt rate regime and learning to label those regions correctly. Then, as a second step, the DAE network learns to effectively remove the 'noise' from this output, thereby converting labels to the desired melt rate output. Thus, although the final output layer of MELTNET is the same as the one that would be used if this were treated as a regression task, the way in which melt rate prediction task is handled by the network is very different. This re-framing of the task of emulating a physical system as a segmentation task has not, to our knowledge, been done before. We find that in this case it outperforms other approaches
but undoubtedly will not be suitable for all problems of this kind.

**Table F1.** parameters for the two intermediate complexity melt rate models (PICO and PLUME), showing both the originally published values and the optimised values that minimised the NRMSE to NEMO melt rates, as described in the Methods.

| Parameter | Model | Value | Original |
|---|---|---|---|
| $\rho_i$ | all | 917 | |
| $\rho_w$ | all | 1030 | |
| $\gamma_T^*$ | PICO | 0.97e6 | 2.00e-5 m s$^{-1}$ |
| $C$ | PICO | 1.00e6 | 0.70e6 m$^6$s$^{-1}$kg$^{-1}$ |
| $\Gamma_{TS}$ | PLUME | 2.19e-4 | 6.00e-4 |
| $E_0$ | PLUME | 1.98e-2 | 3.60e-2 |

## Appendix F: Supplementary Tables

*Author contributions.* S.R. conceived the study, created the synthetic training sets and developed the deep learning methodology. C. B. setup and ran the ocean modelling component. All authors contributed ideas to analysis of the results and to the writing of the manuscript.

*Competing interests.* The contact author has declared that neither they nor their co-authors have any competing interests.

*Acknowledgements.* This work used the ARCHER2 UK National Supercomputing Service (https://www.archer2.ac.uk). S.R. is supported by the PROPHET project, a component of the International Thwaites Glacier Collaboration (ITGC). Support from National Science Foundation (NSF: Grant 1739031) and Natural Environment Research Council (NERC: Grants NE/S006745/1 and NE/S006796/1). C.B. is supported by the European Union's Horizon 2020 research and innovation programme under grant agreement no. 820575 (TiPACCs). We are very grateful to Adrian Jenkins for numerous helpful discussions.

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
