# Peer review of "Predicting ocean-induced ice-shelf melt rates using deep learning"

_The Cryosphere, 2021_

## Author Response (AR1)

We thank all three reviewers for taking the time to read through our paper 'Predicting ocean-induced ice-shelf melt rates using a machine learning image segmentation approach 'and for their thoughtful comments.

The key criticism of two of the reviewers is our use of two networks, consisting of a segmentation network followed by an auto-encoder, instead of a single Convolutional Neural Network (CNN). We explain our reasons for using this approach in more detail below, but fundamentally we must use this two-step approach (or something similar) to arrive at a decision for each pixel of our image. Using the CNN approach as the reviewers suggest, would not give use the required spatially distributed field and its multi-scale dependency on the input fields. Our approach is quite novel and new, and we do realize that we should have spend more space in the manuscript explaining the advantages of our new approach, and had we done so the reviewers would have been in a better position to understand that a typical CNN approach does provide the information required. We will clearly need to expand the discussion of the methodology in our manuscript, but we remain convinced of the methodological advantages of our approach.

In our response, we have first combined these comments and replied to them together. The remainder of our response is structured by replying (in italics) to each reviewer comment sequentially. Where we have made changes these are described in underlined text.

**Extracts from reviews:**

**Regarding choice of architecture (Guillaume Jouvet)**

**As you design an ANN mapping 2D to 2D fields with continuous variables, the most logical and intuitive to me would be to use a standard Convolutional Neural Network (CNN) trained as a regression problem with a L1 or L2 loss (e.g. similarly to the CNN I use to learn ice dynamics). You may have also considered a U-NET architecture as well to better capture underlying multiscales, if any. Therefore, my main point is: why do you split in two networks? -- a first segmentation/classification and an auto-encoder (AE) -- I just do not see what this brings except unnecessary complications (and probable loss of information!). Unfortunately, I could not find any line of justification for this choice, namely transforming the problem into a classification one, and then afterwards recovering the lost information (or 'corrupted' as you term it) by an AE. To me, the final paper should either i) try to simplify their approach using a single and simple regression network if this proves to be as efficient OR ii) clearly justify the choice of going to a more complex network sequence and explain why the simplest approach was unsatisfactory. In case of i), consider revising the paper title and removing references to segmentation.**

**Regarding choice of architecture (Jordi Bolibar)**

**Despite all this strong aspects, I must say I am very surprised by the modelling choices regarding the neural network(s). Right from the title it is clear that the authors have decided to use an image classification architecture for this problem. This is quite bizarre, since this problem is clearly a regression problem, not a classification/segmentation one. As I was reading the manuscript, I was**

expecting the authors to explain such a strange choice, but I could not find any justification for that choice.

This has been by far the most striking feature of this manuscript. The authors have chosen to use an image classification network for a regression problem. From the results, this method seems to work, but in order to do so the authors have had to "force" the architecture into this problem, resulting in some awkward modelling strategies. Since there is no justification for this choice, there might be two potential explanations to this: (1) The authors have a clear strategy behind this, but did not explain it in the main manuscript. (2) The authors have re-used an already existing architecture (a very common and totally correct ML practice) from image classification, and tried to apply it to this problem without knowing that it was designed for a completely different task. I would like to know the exact reasons behind this choice, but on the meantime I will try to argue why I think such an architecture is not the best choice for this task.

Deep learning models can be applied to two main different types of problems: classification and regression. Classification is the most popular one, involving a nonlinear transformation of input data into a new space, in which a segmentation is performed based on a specific number of classes or labels in a supervised or unsupervised manner. On the other hand, regression models are in general less well known, and they are more challenging to train, validate and apply to physical systems. While the validation of a classification model is quite straightforward, since it is very easy to verify if the labels are accurate or not, this is not true for a regression problem. Regression problems for physical systems are trickier to validate, as one needs to make sure that the model is learning the physical relationships for the right reasons.

The fact that the authors chose a classification model for a regression problem has a series of consequences which add unnecessary complexity to the modelling framework:

• The discussion on the choice and impact of the number of classes for the first network could have been completely avoided just by choosing a regression model. Since the modelled variable (melt) is a continuous variable, it does not make sense to model it in a discrete way with a classification framework.

• The authors compensate this strange choice by adding a second neural network, an autoencoder, in order to interpolate the discrete classes obtained by the first network. As for the previous point, this second network could have been directly discarded if a regression model had been chosen.

• The model(s) presented in this study do seem to work, but I cannot help wondering how simpler and potentially faster might have been a solution with a regression network.

Since everything else in the study is well conceived, the model seems to work and the authors have even verified the physical plausibility of the learnt model, I do not think these reasons above are enough to deny publication. However, I would ask the authors at the very least to clearly explain in the discussion the reasoning behind this strange choice and comment on what the use of a regression model could imply for such a modelling framework.

Moreover, if the authors think it is something relatively simple to achieve, I would encourage them to re-train a regression CNN to see if the results are improved. In lines 357-359, the authors mention that they trained a single CNN that performed the same tasks as both networks. If that is really the case, that should be a regression network, otherwise it would not be possible to go from a continuous input to a continuous output. They also mentioned that such a network proved

**harder to train. It would be interesting to know if that is because they simply re-used the same architecture with some minor changes (e.g. just changing some activation functions), or if they chose a specific architecture suited to regression problems. As I said, training and validating regression networks is often trickier, but it is very likely that this might result in a better model. I will not enforce these changes, due to the above mentioned reasons. If they decide that it is too much work and they would rather keep the current model, then this should be clearly added in the discussion as a future perspective, including the current shortcomings of the model. The current model is overly complicated for this problem. A regression model would largely simplify the modelling pipeline, and could potentially result in a more accurate and expressive model.**

**Our response regarding the choice of architecture**

*The main concern raised by reviewers Jordi Bolibar and Guillaume Jouvet was regarding the choice of network architecture used to predict melt rates. The approach we present in the paper is to combine two networks: a segmentation network that takes input images and returns labelled melt rates and an autoencoder that takes these labelled melt rates and converts them to a continuous melt rate field. Jordi and Guillaume question why we did not use a simpler network, for example a CNN or regression network.*

*The motivation behind our choice of a segmentation network is to extract information from our input fields at multiple spatial scales. A typical CNN architecture is made up of several convolutional layers with the final layer being a fully connected network. Each convolutional layer learns the features in the image from the previous layer. The convolutional layer learns to detect features from the incoming image, and this is repeated in every layer until the final layer which typically takes on the role of decision maker, for example, to classify the image as a cat or dog. The convolutional layers perform feature extraction while the last layer is the decision maker. As the input image is passed from layer to layer in the CNN, the information in the image tends to be progressively more abstract and sparser such that some information is lost at the end. Most common CNNs only give a single output as a global decision for the whole input image. In other words, they generally do not make a decision for every single pixel of the image.*

*The segmentation network (segnet) overcomes both limitations. As the input image passes through the convolutional layers, the features are firstly extracted at different spatial scales and therefore, the extracted features at every layer become more meaningful, representable and interpretable by humans. In addition, the features from different spatial scales (coming from different convolutional layers) are progressively reconstructed at the decoder stage. This effectively enables the segnet to learn and internalise a multiscale spatial representation of the inputs representing bathymetry, ice shelf draft, temperature and salinity. The last layer of Segnet classifies every pixel of the penultimate layer s features into one-of-N labels.*

*While a CNN is not limited to classifying images, for example the model by Jouvet (2021), they are limited to operating on the same region of an image, defined by the filter size. Here, therefore, is the great advantage of using a segmentation network: the CNN filters in the segmentation network architecture operate at multiple spatial scales.*

*The magnitude and spatial distribution of melt rates beneath an ice shelf is inherently a problem determined by processes at multiple spatial scales. Some are entirely local: for example the difference between the local melting point and ocean temperature, or the local ice shelf slope;*

*whereas some are nonlocal: for example the path that meltwater plumes take from deep grounding line points, and overall circulation beneath the ice shelf. Thus, our approach makes use of the multiscale learning enabled by the segmentation network architecture to predict melt rates using local and nonlocal features in our input images.*

*The downside of our approach is that, since the segmentation network is designed to output labelled images, we then need to convert these labels to a continuous melt rate field in order to have a useful output that could be deployed directly within an ice sheet model. In practice this is only a minor inconvenience since the output from the first network serves directly as the input for the second network.*

*There are also clear advantages to having two distinct networks. The segmentation network ultimately does the heavy lifting, extracting features in an image at various spatial scales to predict where there might be strong ice-shelf melting, some refreezing, etc., which it assigns as labelled regions of the image. The autoencoder network, with an architecture typically used for denoising an image, only needs to learn how to convert from labels to a continuous field. An autoencoder is very suitable for this since the problem is somewhat analogous to taking a compressed pixelated representation of an image and extracting the original. In this context, the  compression  is that melt rates are only represented by N labels. Each network specialises in its respective task, and, for example, the second step goes some way to  regularise  the output melt rate field, since part of what the autoencoder learns is that the melt rate field is generally quite smooth. Essentially, the autoencoder interpolates between the discrete labels from the segmentation network so that a generally smooth map is achieved. This also enables this map to be more interpretable to human users and more useful for input into an ice sheet model.*

*We accept that justification for our choice of architecture is largely missing from the manuscript, and we will add a condensed explanation of this in a revised version.*

*Jordi Bolivar states that our choice of architecture is 'quite bizarre, since this problem is clearly a regression problem, not a classification/segmentation one'. From a machine learning point of view, prediction is an extrapolation problem, and not interpolation. If we treated this as a regression problem there would indeed be no need for the second autoencoder, and the model becomes an end-to-end model. However, such an end-to-end model would struggle to learn the extrapolation problem and is known to have poor generalization ability in the event where it is presented with data different from what it has been trained on. Furthermore, as mentioned by Jordi, regression problems for physical systems are trickier to validate, as one needs to make sure that the model is learning the physical relationships for the right reasons. So, in an end-to-end model, there is no guarantee that the model is learning the physical relationships. One of the novel aspects of our methodology is that we are able to circumvent this problem by casting the prediction problem as a sequence of a classification problem followed by interpolation problem.*

*Jordi Bolibar references a section in our manuscript regarding an alternative architecture that we tried to use to avoid having two distinct networks.  The line numbers he refers to appear incorrect, but we assume he is referring to lines 337-339:  We developed and tested an architecture that combined both DAE and segmentation components into one network, however this proved harder to optimise than the approach we have presented here.  As Jordi Bolibar explains, this network is solving a regression rather than classification problem, since its output is a continuous melt rate field, although it is not a CNN network.  Once again, we accept that we did not go into sufficient details in our manuscript about alternative architectures that we tried for this problem, and we can provide more detail on this point in a revised manuscript. As we stated in the manuscript, this architecture proved much harder to train. This is not surprising, since the Segnet is solving a multiclass*

*classification problem while the DAE is solving an interpolation problem (as a nonlinear regressor). It is tempting to combine both networks into a single integrated network, since the architectures are visually similar, however each network solves a different problem and so any attempt to optimise the single integrated network will only favour one over the other.*

*Guillaume Jouvet specifically mentioned U-NET architecture as an alternative. U-NET is a very similar architecture to the segmentation network that we have used, only differing in how information is transferred from pooling to unpooling, and thus also results in a labelled image. Since the output of U-NET would also need to be converted from labels to a continuous melt rate field, and the main concern seemed to be regarding our choice of using two networks rather than one, we do not understand how this approach would improve on, or differ significantly from, our own. From previous research in related applications (e.g. lung segmentation), it was shown that both Segnet and U-NET show similar accuracy performance but that Segnet is computationally less intensive (Saood & Hatem, 2021)*

We have taken on board the reviewers concern regarding our choice of architecture and further explored alternative methods but our approach continues to perform significantly better than using a single network. We have added justification for our choice of architecture in the discussion section, along with briefly mentioning other options that we explored and our thoughts on why these do not perform as well. We have also considerably expanded the training and validation sets to lend further confidence with regards to the performance of our model.

**Review by Timothy Smith**

**Specific Comments**

**Making predictions with an ML surrogate is much cheaper than running an ocean model, but the training is not free and can be quite computationally demanding. The authors allude to this in Line 64:**

**"Since the computational cost of a machine learning algorithm is insignificant once it has been trained,"**

**However, I think it should be mentioned in this way in the abstract and earlier in the introduction, e.g. around Line 39, since model training can be a major computational expense in ML for high dimensional problems such as those in the geosciences. Moreover, I think the paper would be strengthened by providing some estimate of the computational costs of training and and making predictions with MELTNET. This could be as simple as a table with training and validation walltime for each network, along with the architecture (e.g. was this on a laptop or run in the cloud? how many nodes/cores/threads were used?). Providing these details would help quantify the statement that predictions are almost free, and would help establish to the community that, generally speaking, ML based emulators are worth pursuing.**

**Training of the submitted version of the segmentation network, which was by far the most computationally expensive, took less than 24 hours day on an NVIDIA Quadro K5200 GPU. The much larger computational cost is associated with running the thousands of NEMO simulations to create the training set that the network is trained on. Calculating melt rates from an input once trained takes MELTNET ~0.05 seconds. We will add more information/discussion regarding computational cost in a revised manuscript.**

We have added a sentence in the abstract and introduction emphasising the advantages in terms of computational cost. In addition, we have added a paragraph in the discussion that expands on the computational costs and speedup when compared to the other models used.

**Lines 101-105: I have a philosophical disagreement with using the temperature and salinity conditions at the icefront instead of using the open boundary conditions to NEMO. MELTNET is an emulator for NEMO (as far as I understand), and therefore it should not use anything that NEMO produces as an input. Rather, it should be given the same boundary conditions and then bypass NEMO altogether. It is fortunate and also useful to note that using either of these conditions provides essentially equivalent result, since this provides an exciting opportunity in the case where ample icefront T/S data are available and could be used as inputs to MELTNET. However, in this paper MELTNET is being presented as a NEMO emulator, so I recommend keeping this note, but using results based on using the same forcing for NEMO and for MELTNET.**

*The reviewer raises an excellent point and we did discuss this very same point at length during our experimental design phase. There are presumably no simple answers here, but as the reviewer points out, our results are insensitive to these choices. We agree with the reviewer that if MELTNET is considered an emulator of NEMO then it should not be given anything from NEMO as an input, and we were initially reluctant to do this in case it provided an unfair advantage. Due to the idealised nature of the ocean modelling setup (e.g. no surface forcing), we think there is little water mass transformation occurring between the northern boundary and the ice front. For this reason, the results were fairly insensitive to this choice. The reason we ended up using ice front NEMO T+S as input to the network was that this would make MELTNET a more useful tool for ice sheet modellers moving forward and is more consistent with other parameterisations (Burgard et al. 2022). In terms of future climate forcing – open ocean T+S conditions are unlikely to change markedly around Antarctica, and instead the main driver of change is expected to be processes such as a deepening of the thermocline near and beneath ice shelves, enabling warmer water to access the ice shelf base. Therefore, we believe ice sheet modellers would find it more useful to be able to force MELTNET with ice front conditions than open ocean conditions.*

We have reworked the section on input temperature and salinity in a way that hopefully makes our motivation behind this choice clearer.

**Lines 262-270 and 309-314: The comparison to PICO and PLUME in this paper is entirely appropriate. However, in some sense the comparison is unfair since these models are calibrated by tuning 2 global parameters while MELTNET has many degrees of freedom which are optimized during training. One could argue that to make the comparison as fair as possible PICO and PLUME should have spatially varying parameters, which should be calibrated. PICO and PLUME are not used in this way, so I don't think this should be implemented, but it raises a couple of points that I think are worth discussing.**

- **Some details on MELTNET should be included, such as: the degrees of freedom (i.e. number of nodes) for each layer, the number of layers in each stage of the model, and the cost function that is optimized during training (is it simply the norm of the model/data misfit? is there regularization used to penalize large weights?)**

We agree that these important details were missing and we have added extensive additional information on these points in two appendices. We chose not to include these in the main paper since the majority of the readership will likely be unfamiliar with the terminology and uninterested in some of these details.

- **The PICO and PLUME parameters are not really optimized, but "hand tuned" so I would suggest changing that wording, especially since MELTNET *is* optimized (trained). Additionally, I think that the difference in degrees of freedom could be worth**

**mentioning. In a sense, one could make the argument that the neural network is a way of capturing the additional degrees of freedom that we would want to have in the PICO or PLUME models (for instance with spatially varying parameter fields) but that we don't know how to specify. All in all - I think some discussion or hypotheses for potential reasons on why MELTNET outperforms these models would improve the paper.**

We admittedly skip over the details of this in the submitted manuscript but the PICO and PLUME parameters are optimised – we use MATLAB's fmincon optimisation function to find the parameters for each model that minimise our chosen cost function. We have added this information to the relevant section and we also added a section in the discussion reflecting on the reviewers second point.

**Minor/Technical Comments and Suggestions**

**lines 35 and 37: Please fix citations: "e.g." comes after the citation but should come before**

Done

**line 59: "lower complexity parameterizations", I suggest to make the minor clarification that these are ice sheet model parameterizations**

Done

**Fig 1: I suggest adding a note in the figure caption mentioning that the GAN step is merely a method to generate many realistic T/S profiles for training, but is not necessary for making predictions once MELTNET is trained, with a reference to section 2.3.2 (and possibly Appendix A). The training and prediction stages are clearly delineated in the figure, and your figure caption is well written, but I think adding a note like this will help a reader who is skimming through the figures as quickly as possible (which, of course, will be many people …).**

Done

**line 120: I recommend being a bit more specific than "these filters are learned", for instance something like "the weights that make up these filters are learned"**

Done

**line 148: I recommend referencing Fig B2 before making the parenthetical note comparing swish are ReLU, since I went to Fig B2 looking for a comparison between the two, rather than a description of the normalisation and layers**

This was resolved by moving the explanation of the choice of activation function to the detailed model description in the appendix.

**line 159-161: For the ocean modellers, could you provide some citations for the specific subgrid-scale parameterizations used? E.g. it sounds like the vertical mixing scheme is from (Gaspar et al, 1990, https://doi.org/10.1029/JC095iC09p16179). What is the scheme for generating lateral viscosity coefficients (Smagorinsky, Leith, etc)? What horizontal and vertical diffusivities are used? I think these details will be nice without being overloading.**

**Line 169: what is the vertical spacing for each of the 45 vertical levels?**

*Each vertical level had a vertical spacing of ~45.5m.*

**Line 208: wouldn't the constraint on ice shelf area be a maximum, rather than a minimum?**

*The idea of this constraint is to ensure that smaller ice shelves are sometimes generated, even if the geometry would have been accepted by the algorithm otherwise because it fulfils all other criteria.*

**Fig 3: Have these ice shelf images been rotated to all be in the same orientation, since you mention that you provide ice shelves in all cardinal directions to MELTNET? If so, you may want to mention that some of these are rotated (e.g. "north isn't always up") in the caption.**

Originally this was the case but our latest version of MELTNET actually no longer gets trained on rotated images and we keep the orientation of the images the same as NEMO. Since Coriolis is included in the ocean model, melt rates are not independent of orientation and removing this aspect of data augmentation lead to improvements in MELTNET performance

**Line 242: Please add the year to the citation Boyer et al**

Done

**Line 250: Do you mean to say "all models *are* judged"?**

Fixed

**Line 260: Misspelled "parameterisations"**

Fixed

**Line 264: "tunetable" -> "tunable"**

Fixed

**Fig 4 caption: Please add the units to the contour intervals in the statement: "contoured from 200 to 800 m at 100 *m* intervals"**

Done

**Line 291: I would recommend putting this part of the paper (discussing the idealized geometry experiments) in its own subsection or even section. This would help the reader since you are testing a new hypothesis.**

Done

**Line 351: Another small wording suggestion: "NN emulators *that are* constrained, ..."**

Done

**Review by Jordi Bolivar**

*Note: major comments related to network architecture were extracted to be addressed separately above.*

**1.2**

Another aspect of the modelling framework that I believe should be improved is its validation. According to the manuscript, only 5% of the dataset is used for validation, which seems extremely low. The authors justify this low fraction of data for test arguing that this maximizes the training dataset, thus improving the overall model performance. This is even more surprising knowing that this is in fact a surrogate model, whose training and validation data can be generated at will. Expanding the validation dataset would be as easy as generating more synthetic ice-shelf geometries and running NEMO on them. From Figure B4 we can see that the train performance plateaus at around 2500 synthetic cases. However, there is no information on how the test set impacts the performance. In machine learning it is essential to monitor the simultaneous evolution of the train and test performance, since they give important clues regarding overfitting or underfitting.

Some extra analyses should be performed in order to improve our confidence in the surrogate model(s):

• I believe the test dataset should be expanded. 5% might (or will likely) not be enough to correctly evaluate the out-of-sample model performance in a large variety of ice-shelf and ocean configurations.

• The test performance should be added to Figure B4, in order to track its evolution with different dataset sizes. If computational costs are behind the use of just 5%, I would still encourage the authors to expand it as much as possible, and then add these reasons explicitly in the manuscript.

It is correct that, since MELTNET is trained using synthetic data, we are not limited by the size of our training set - except in terms of the computational cost required to run the NEMO simulations (which is not insignificant). However, perhaps there is some confusion because the accuracy plotted in Figure B4 is validation accuracy, not training accuracy. We can add a line showing training accuracy in the same plot and we can expand the size of the overall training set (which would also increase the size of the validation set) to provide further evidence that the size of the validation set is sufficient.

We have substantially increased the size of the training set by a factor of ~4, and the validation set, which is now a larger fraction of the overall samples, has increased by a factor of ~8. We have also included a more informative plot showing replacing the one in Figure B4.

**1.3**

Another downside of the manuscript is the lack of transparency regarding the model details. The main issue in my opinion is the fact that the model source code is not open-source. There is only a statement saying that the synthetic geometries are available upon request, without any mention of the model code itself. This makes it even harder to review the model, and goes against the open science values from journals such as The Cryosphere. Many of my doubts or questions could have been directly resolved by checking a properly documented repository on GitHub (or elsewhere). Therefore, I strongly encourage the authors to share their source code in a public repository. By making it citeable (e.g. using Zenodo), there are virtually no downsides to sharing it.

We have added a GitHub repository with the model code, a link can be found in the manuscript code availability section

**This has also been commented by the other reviewer. I think overall there is a lack of details regarding the model configuration in the manuscript. I understand that the authors do not want to flood the text with technicalities, but it would still be interesting to know a little bit more about the model in an Appendix or Supplementary material. Details regarding the optimizer for the gradient descent, regularization techniques used to avoid overfitting, learning rates, etc...**

We have added extensive additional information on these points in two appendices, for both the segmentation network and the DAE network

**2.0 Specific comments**

**L120 Please add more details about the optimizer and gradient descent in either the text or an additional section in the Appendix or Supplementary material.**

We have added considerable detail on this in the appendix.

**L126-127 By simply evaluating the loss at the pixels covering the ice shelf this could be easily solved. A matrix mask could be used to filter out those values. This yet another consequence of using a classification framework.**

We use a weighted loss function, based on label frequency but with zero weight attached to pixels that lie outside of the ice shelf, which is essentially the same as directly masking out those pixels. This is now described in more detail in the appendix.

**L133 A simple leaky ReLu could have sufficed, which is also less computationally expensive**

*We tested leaky ReLu and found that the swish activation function outperformed it. The Swish activation function is bounded below (meaning as x approaches negative infinity, y approaches some constant value) but unbounded above (meaning as x approaches positive infinity, y approaches infinity). However, unlike leaky ReLU, Swish is smooth (it does not have sudden changes of motion or a vertex). Unboundedness is desirable for activation functions because it avoids a slow training time during near-zero gradients — functions like sigmoid or tanh are bounded above and below, so the network needs to be carefully initialized to stay within the limitations of these functions. Being bounded below may be advantageous because of strong regularization — functions that approach zero in a limit to negative infinity are great at regularization because large negative inputs are discarded. This is important at the beginning of training when large negative activation inputs are common. In addition, Swish's non-monotonicity increases 'expressivity' of an input and improves gradient flow. The smoothness helps optimise and generalise the performance of Segnet and autoencoder.*

We have added some justification for this choice in the appendices that describe the networks in detail

**L277-280 This should be explained in the legend, otherwise it is impossible to understand.**

This has been added to the figure caption

**L281 By remaining panels do you mean the panels shown in Fig. 4?**

Yes, modified the sentence to make this clearer

**L283-284 This should also be mentioned in the figure. It is important to mention that you are showing an out-of-sample performance.**

Done

**It is also unclear why the performances of the two parametrizations are not included. One would expect to see the comparison here, otherwise there is no baseline performance to compare with.**

*The purpose of this plot is to show visually how the sampling from distribution of scores works out in practice. A comparison to performance of the two parameterisations is shown in Figure 5.*

**Figure 4 "Note the colour map gradient is not linear, but is greatest around zero, to make it easier to distinguish the magnitude of melting/refreezing over the bulk of the ice shelves." What do you mean? To me the colourmap from the plot appears to be linear.**

*The colour axis interval is constant but the colourmap gradient is not linear, for the reason stated.*

**L303-304 Couldn't you change the loss of these two models? This could be easily solved by tuning all models with a combined loss: e.g. the (NRMSE local + NRMSE average)/2.**

*That certainly could have been done but as explained elsewhere in the paper average melt rate is less important than matching the melt rate in certain key parts of the ice shelf and so tuning for average melt would not make much sense.*

**L315 Indeed, this study has focused on modelling the spatial information of ice shelf melt. Modelling of the temporal dimension remains untackled, and it might prove more challenging to do (see e.g. Bolibar et al. (2020) The Cryosphere). A validation in the spatial dimension doesn't ensure a good performance in the temporal dimension, which would be mandatory for any real world application as a surrogate for NEMO.**

*We do not see this as a major limitation of the model we present. In the context of Antarctic ice shelves, melt rates vary as a function of ice shelf geometry and ocean forcing, both of which will vary as a function of time, and both of which are inputs to the network. Therefore, assuming the system has had sufficient time to adjust to any changes in forcing, we should expect the network to capture this temporal dimension. Variability at very short timescales, for which the cavity circulation has not had time to adjust to, is unlikely to be relevant to large-scale ice sheet simulations.*

**L331 Do you mean to the surrogate model? How would you add new physical processes to a surrogate model? I am not sure this is that straightforward to achieve. This model acts as a black box here, it just can be trusted because it is emulating a physical model that can be well understood.**

New physical processes can be incorporated into the network automatically if they are included in the ocean model that network is trained to emulate. We have expanded on this point slightly in the discussion.

**Review by Guillaume Jouvet**

*Note: major comments related to network architecture were extracted to be addressed separately above.*

**The most convincing to me is the fidelity result of the ANN to the instructor NEMO, but then I think it would be good to clearly give clear numbers and report it in abstract and conclusion. You**

may choose a metric and state how far (in %) is MELTNET solutions from NEMO? By contrast, I unsure that comparisons with other simpler models should be too elaborated. E.g. Fig. 4 is useful as it shows that the loss in accuracy between MELTNET from NEMO is small/negligible compared to the discrepancy between low and high complexity models (PICO vs PLUME). I think that is enough as I expect the paper mostly to focus on the accuracy of the MELTNET to reproduce its instructor model -- the in-between model comparisons being a substantial task to make sure this is done fairly (I don't have the expertise to assess this). From Fig. 4 I retain that comparing MELTNET with other models is roughly the same as comparing NEMO with others as the two are (hopefully) very close to each other (as the ANN makes a very good job). This also means that the rest is a pure comparison of models no longer involving deep learning, and this may go beyond this scope of the paper. In conclusion, I would probably keep the comparison with PICO & PLUME rather concise, and favor MELTNET/NEMO comparisons.**

*We agree that it would be helpful to provide a metric in the abstract that directly compares the performance of MELTNET with NEMO, as this will be one of the things readers will want to know from the outset. We would prefer to keep our comparison with the PICO and PLUME parameterisations, however, since these are commonly used by the ice sheet modelling community and so a detailed comparison is the most likely approach to convince modellers that MELTNET provides a useful alternative.*

We have added a sentence in the abstract that gives a metric for the ability of MELTNET to replicate NEMO melt rates.

**The main point of using deep learning emulators is the huge computational gain versus minor loss in accuracy. While you have quantified the accuracy (Fig. 4), it is a pity that you do not do it for computational time. What speed-up? I expect several orders of magnitude. Quantifying the computational time is essential for your paper. You may also comment on the fact ANNs run extremely well on GPU (which is not the case of CPU), giving another important advantage of your method (compared e.g., to NEMO which may not take the same advantage on GPU).**

We have added a section in the discussion that goes into this in some detail and we also mention this in the abstract

**I think the paper can be made more efficient by moving technical machinery in Section 2.3.1 to appendix. The generation of synthetic geometries is necessary, but of lower interest. Moreover, using a GAN is an elegant strategy, but this is probably nonessential.**

We have moved section 2.3.1 to the appendix, as suggested, although information regarding the GAN is already almost all in the appendix and we feel what is included in the main text is a bare minimum needed to understand our approach.

**I think Section 2.2 should come first for the sake of clarity. It sounds more logical to first describe the physical model, and then the ANN you design to learn from the physical model as the choice of the ANN architecture is motivated by the type of emulated physics.**

We have moved section 2.2 to the start of the methods, as suggested.

**Why not using Antarctica and Greenland real topographies to generate ice shelf geometry? This would avoid to generate synthetic geometries?**

*As we state in the paper, there are insufficient real world examples of ice shelves to make a useful training set for our machine learning approach. In particular since we choose to predict melt rates beneath entire ice shelves, rather than subsets of ice shelves. As explained in our response to questions about the choice of architecture, this is an intentional choice to allow for the possibility of the network to make predictions based on non-local general circulation processes rather than just local melt processes.*

**Minor comments:**

**For clarity, I think you should call MELTNET like NEMO-trained MELTNET or at least include NEMO as you may train MELTNET with other models.**

*It is certainly true that the approach we have used is not limited to being trained by NEMO, and we can add a comment regarding this in the paper, however we prefer to keep the name of the model concise and since only one version of the model exists we do not feel that it is necessary to specify in its name which ocean model the network has been trained on.*

**l76: suggest "the inputs and NEMO resulting melt rates …"**

Done

**l79: fix the typo with "paramterisations"**

Done

**"These filters are learnt ..." not sure this is understandable for who is unfamiliar with ML vocabulary**

Rephrased this sentence

**l127-128: why not cropping the area of interest instead of weighting? Anyway, you normally can feed your ANN with different frame dimensions.**

Since the ice shelves have a wide variety of shapes and sizes, cropping would still result in many training images with far more pixels of some classes than others. Weighting is a very common approach for image segmentation used to ensure that this does not bias the training to certain classes over other, less represented classes.

**l129: You can state clearly you do data augmentation.**

Our latest version of MELTNET actually no longer gets trained on rotated images and we keep the orientation of the images the same as NEMO. Since Coriolis is included in the ocean model, melt rates are not independent of orientation and removing this aspect of data augmentation lead to improvements in MELTNET performance.

**l169 2000 m**

Done

**l273 Figure 2 presents THE main result of the study, ….**

Done

**Is the AE a U-NET architecture? If yes, you should say it.**

No this is a very different architecture, as with the segmentation network we have expanded on the description in the appendix

**l 352: you may call it PINN for Physically Informed Neural Network.**

We are not sure what part the reviewers suggestion refers to

**l 345: $\sim 502$ km$^2$**

Done

**References**

Saood, A., Hatem, I. COVID-19 lung CT image segmentation using deep learning methods: U-Net versus SegNet. *BMC Med Imaging* **21,** 19 (2021). https://doi.org/10.1186/s12880-020-00529-5

Jouvet G, Cordonnier G, Kim B, Lüthi M, Vieli A, Aschwanden A (2021). Deep learning speeds up ice flow modelling by several orders of magnitude. Journal of Glaciology 1–14. https://doi.org/10.1017/jog.2021.120

Burgard, C., Jourdain, N. C., Reese, R., Jenkins, A., and Mathiot, P.: An assessment of basal melt parameterisations for Antarctic ice shelves, The Cryosphere Discuss. [preprint], https://doi.org/10.5194/tc-2022-32, in review, 2022.

---

## Referee Report (RR1)

**Review of "Predicting ocean-induced ice-shelf melt rates using a machine learning image segmentation approach" by Rosier et al.**

The Cryosphere

**Round 2**

**1 General comments**

In this second review I address the new version submitted by the authors, as well as their previous response to the reviewers.

I am happy to see some of the changes suggested in the new version, but I must admit there are still some aspects that remain unclear to me. I will just focus on two main ones, since as I said in my previous review, I think the paper is overall of high quality, with only some methodological shortcomings.

**1.1 GC1: Choice of a classification modelling framework**

I appreciate the extended explanations on this modelling choice, but according to the authors' reply to the reviewers, I am still not sure that we are on the same page in terms of our previous questions.

The authors justify their choice saying that if they wanted to use a regression network, they would have to apply a CNN like the one in Jouvet et al. (2021). I don't understand this comparison between different architectures, since any neural network architecture can be used for both classification and regression. Even U-Net can be used for regression, it would be as simple as changing the very last layer, with a single neuron and a linear activation function, and replacing the loss function for something like RMSE or MAE. What defines a network as a classification or regression one is not its architecture, but its output layer and loss function. It is unclear if the authors have attempted this extremely simple change, or if they have tried other different architectures. Directly adjusting U-Net for a regression problem would the most straightforward and fair way to test this.

I understand that if this current architecture works, it makes sense to keep it. But I'm also worried that such an architecture might end up being used as a reference for this sort of tasks, which is clearly and overcomplication of the problem. If the U-Net and the autoencoder where optimized at the same time (i.e. with a single loss at the output of the autoencoder), this would already make more sense, since backpropagation would take place as a regression

problem. However, destroying information to train a first classification network and then asking another network to undo the damage looks problematic to me.

In order to avoid a sterile debate over this, I ask the authors the following:

- Can you please clearly explain what exact architecture and changes you applied to the U-Net in order to use it as a regression problem? This should be clearly explained in the Appendix. I would strongly encourage the authors to test the very simple changes proposed above to use the U-Net for regression. Simply change the output layer and the loss function. This should be less than 5 lines of code.

- You should clearly state in the methods section and the abstract that this is a regression problem. You can explain that this more complex combined method of classification + regression works well here, but the reader should know that this is an exception, not the norm.

**1.2   GC2: Absence of test dataset**

One aspect that I admit escaped my first review is the absence of a test dataset. Currently the authors are splitting their total dataset into 90% training and 10% validation. Then, they tune the hyperparameters of the networks based on the results of the validation dataset, but no independent assessment of model performance is made. This is highly problematic, since the model hyperparameters are overfitting the validation dataset, which is also used to assess the final model performance.

The usual and recommended practice in machine learning is to split the dataset into train, validation and test datasets. The reasons to use an independent test dataset are widely explained in the literature, so I leave a reference here for more details (see points 2.2 and 3.1): Lones (2022), How to avoid machine learning pitfalls: a guide for academic researchers.

Before publication, the authors should repeat the training process, putting aside a test dataset. A possible division could be 70% training, 20% validation and 10% test. The authors are free to choose these ratios, but I would encourage them to use at least 10% on test. The split between these 3 datasets should be done before training, and the test dataset should be kept aside and not used until the very last moment, which will serve to have the "real" model performance. The final performance on the test dataset should be at least slightly lower than the one on the validation dataset (if the model is generalizing correctly). All figures displaying model performance should be updated with the results of the test dataset.

In order to allow a fair comparison against PICO and PLUME, these dataset divisions should also be applied throughout all models (particularly for Fig 4).

---

## Author Response (AR2)

We thank the reviewers for taking the time to conduct a second review of our manuscript. We have compiled comments from both reviewers below (in bold) together with our response and actions taken (in italics).

**I appreciate the extended explanations on this modelling choice, but according to the authors' reply to the reviewers, I am still not sure that we are on the same page in terms of our previous questions. The authors justify their choice saying that if they wanted to use a regression network, they would have to apply a CNN like the one in Jouvet et al. (2021). I don't understand this comparison between different architectures, since any neural network architecture can be used for both classification and regression. Even U-Net can be used for regression, it would be as simple as changing the very last layer, with a single neuron and a linear activation function, and replacing the loss function for something like RMSE or MAE. What defines a network as a classification or regression one is not its architecture, but its output layer and loss function. It is unclear if the authors have attempted this extremely simple change, or if they have tried other different architectures. Directly adjusting U-Net for a regression problem would the most straightforward and fair way to test this. I understand that if this current architecture works, it makes sense to keep it. But I'm also worried that such an architecture might end up being used as a reference for this sort of tasks, which is clearly and overcomplication of the problem. If the U-Net and the autoencoder where optimized at the same time (i.e. with a single loss at the output of the autoencoder), this would already make more sense, since backpropagation would take place as a regression problem. However, destroying information to train a first classification network and then asking another network to undo the damage looks problematic to me. In order to avoid a sterile debate over this, I ask the authors the following:**

**• Can you please clearly explain what exact architecture and changes you applied to the U-Net in order to use it as a regression problem? This should be clearly explained in the Appendix. I would strongly encourage the authors to test the very simple changes proposed above to use the U-Net for regression. Simply change the output layer and the loss function. This should be less than 5 lines of code.**

**• You should clearly state in the methods section and the abstract that this is a regression problem. You can explain that this more complex combined method of classification + regression works well here, but the reader should know that this is an exception, not the norm.**

*We accept that in our previous review we did not unequivocally address the reviewers concerns with regards to our choice of network architecture and we trust that the additional details provided in the manuscript and extended testing can resolve this. We tested many architectures during the development of MELTNET and in the manuscript we have added a direct comparison with two alternatives, including the exact approach suggested by the reviewer in their most recent comments. As we now show in detail in a new appendix and figure, MELTNET clearly outperforms both a CNN-type regression architecture and a regression version of the UNET-type architecture that we use within MELTNET.*

*We have a different perspective to the reviewer on the task that MELTNET is solving. The reviewer continues to refer to this as a regression problem and instead we view our method as performing a segmentation task (the heavy lifting of prediction, done using a variant of RESUNET), followed by denoising (where the low resolution blocky nature of the melt rate labels is converted to highly resolved melt rates, done using a denoising autoencoder). In this way, the approach is very different from treating the problem directly as a regression task, with both advantages and disadvantages. In this case, it seems that the advantages outweigh the disadvantages, and the performance is*

*consistently better than directly solving a regression problem. Thus, the approach is not a roundabout and long-winded regression, but something different. If it were simply an overcomplicated way of doing regression then we agree with the reviewer, information is lost in the process and so surely there is a better way, and yet here our method performs better - because it is doing something entirely different.*

*Ultimately, we do not believe the choice of deep learning architecture should be the focus of this paper and our aim is not to show that splitting this melt rate prediction task into two networks is a silver bullet for other similar problems. For this reason, we have changed the title of our paper to avoid any confusion that may have arisen on the paper's goal. That being said, we do not understand the reviewer's insistence that this method is a clear overcomplication and their apparent concern that our approach might be tried in other contexts. The statement that a combination of classification and regression works well here being an exception rather than the norm is curious since to our knowledge the reviewer has not tested this approach themselves. Both before and since receiving reviews on our initial manuscript submission we have extensively tested alternatives and cannot find anything that outperforms the methodology that we present in the paper. Therefore, while we do make it clear that this is a novel method that has not been used before, we see no justification to specifically discourage the reader from attempting to use a similar approach for their own work. The field of deep learning is evolving rapidly and no doubt an architecture exists or will be developed soon that outperforms the method presented in the paper, but that has no bearing on our results or conclusions.*

**One aspect that I admit escaped my first review is the absence of a test dataset. Currently the authors are splitting their total dataset into 90% training and 10% validation. Then, they tune the hyperparameters of the networks based on the results of the validation dataset, but no independent assessment of model performance is made. This is highly problematic, since the model hyperparameters are overfitting the validation dataset, which is also used to assess the final model performance. The usual and recommended practice in machine learning is to split the dataset into train, validation and test datasets. The reasons to use an independent test dataset are widely explained in the literature, so I leave a reference here for more details (see points 2.2 and 3.1): Lones (2022), How to avoid machine learning pitfalls: a guide for academic researchers. Before publication, the authors should repeat the training process, putting aside a test dataset. A possible division could be 70% training, 20% validation and 10% test. The authors are free to choose these ratios, but I would encourage them to use at least 10% on test. The split between these 3 datasets should be done before training, and the test dataset should be kept aside and not used until the very last moment, which will serve to have the "real" model performance. The final performance on the test dataset should be at least slightly lower than the one on the validation dataset (if the model is generalizing correctly). All figures displaying model performance should be updated with the results of the test dataset. In order to allow a fair comparison against PICO and PLUME, these dataset divisions should also be applied throughout all models (particularly for Fig 4).**

*To address this criticism we have conducted a further ~1000 NEMO simulations, comprising approximately 10% of our total data, that is used as our 'test' set as suggested by the reviewer. These simulations are not seen during training, nor were they used to optimise model hyperparameters. We now use this test dataset, rather than the previously defined validation set, to evaluate the performance of MELTNET, together with the PICO and PLUME parameterisations. We have updated all relevant figures based on this new test set. We have also clarified this split in our*

*dataset wherever relevant in the text. Our finding shows that this change has no discernible impact on the model performance and does not affect any of the conclusions of our paper.*

**- Line 92: Please cite the GGL90 scheme - I'm assuming this is what is being used (Gaspar et al, 1990, https://doi.org/10.1029/JC095iC09p16179)**

*The turbulent kinetic energy scheme is indeed based on Gaspar et al, (1990) but has since been heavily modified, see p54 of Madec et al. (1998) or 9.1.3 in Madec et al. (2019) for details. We have added a reference to Madec et al. (1998).*

**- Line 95: Please cite the Redi scheme, again I'm only assuming this is what's being used. (Redi, 1982, http://journals.ametsoc.org/view/journals/phoc/12/10/1520-0485_1982_012_1154_oimbcr_2_0_co_2.xml) If Gent and McWilliams, (1990, http://journals.ametsoc.org/view/journals/phoc/20/1/1520-0485_1990_020_0150_imiocm_2_0_co_2.xml) is being used please cite that too.**

*L95 was discussing the three-equation parameterization but we think the reviewer here was asking about subgrid scale eddy parameterisations such as GM. In this instance we did not use a GM-like parameterisation. We do not think this is unusual at these latitudes with eddy permitting resolution.*

**- Line 101: Please either say that the vertical spacing is 45.5m or that the vertical levels are evenly spaced.**

*We have specified that the vertical levels are evenly spaced*

**- Appendix C and Figure C1: I think the manuscript is greatly improved by adding the 2 appendices with details on the neural network architecture - this makes the results more reproducible without being a distraction in the main text. Two very minor suggestions here:**
**1) how big are the convolutional layers used in the Auto Encoder (for inverse classification)? This could be added to the figure or to the text in this appendix.**
**2) the regularization parameter amplitude for this architecture could be added in line 574 for completeness.**

*We have added this information to the manuscript as suggested by the reviewer*